# GRAPHICAL MULTIOUTPUT GAUSSIAN PROCESS WITH ATTENTION

**Yijue Dai, Wenzhong Yan & Feng Yin**[*]
School of Science and Engineering
The Chinese University of Hong Kong, Shenzhen, China
`yinfeng@cuhk.edu.cn, yijuedai@link.cuhk.edu.cn`

## ABSTRACT

Integrating information while recognizing dependence from multiple data sources and enhancing the predictive performance of the multi-output regression are challenging tasks. Multioutput Gaussian Process (MOGP) methods offer outstanding solutions with tractable predictions and uncertainty quantification. However, their practical applications are hindered by high computational complexity and storage demand. Additionally, there exist model mismatches in existing MOGP models when dealing with non-Gaussian data. To improve the model representation ability in terms of flexibility, optimality, and scalability, this paper introduces a novel multi-output regression framework, termed Graphical MOGP (GMOGP), which is empowered by: (i) Generating flexible Gaussian process priors consolidated from identified parents, (ii) providing dependent processes with attention-based graphical representations, and (iii) achieving Pareto optimal solutions of kernel hyperparameters via a distributed learning framework. Numerical results confirm that the proposed GMOGP significantly outperforms state-of-the-art MOGP alternatives in predictive performance, as well as in time and memory efficiency, across various synthetic and real datasets.

## 1 INTRODUCTION

For inference on functions, the Gaussian Process (GP) serves as the most prominent Bayesian non-parametric model, offering not only function values but also predictive distributions. Capturing the input-output relationship with a multivariate Gaussian at finite locations grants tractable predictions and predictive uncertainty (Rasmussen & Williams, 2006). The scenarios where resource constraints impede the collection of ample data samples motivate the development of Multioutput GP (MOGP) models, as the MOGP can correlate data from multiple diverse sources and obtain better predictive accuracy than isolated models. Practical applications of the existing MOGP methods are widespread, such as time-series analysis (Dürichen et al., 2014), water resource forecasting (Pastrana-Cortés et al., 2023), and group-structured data clustering (Leroy et al., 2023).

The popular MOGP models are mainly rooted in the Linear Model of Coregionalization (LMC) proposed by Goulard & Voltz (1992). Modeling all outputs as the linear combinations of shared latent GPs, the LMC correlates the diverse data by a covariance matrix. There also exist other MOGP models, such as (1) transformation-based models that treat outputs as augmented inputs, transforming the MOGP into a sequence of single output GPs (Requeima et al., 2019); (2) discrepancy-based models, which focus on transferring information from low-fidelity outputs to progressively enhance the performance of high-fidelity target outputs (Perdikaris et al., 2017; Requeima et al., 2019). However, the challenges of providing understandable output correlations and a probable output hierarchy have impeded their advancement. To date, improved learning performance of the LMC is achieved by deriving representative covariance functions (Bonilla et al., 2007; Wilson et al., 2012; Dai et al., 2020; Chen et al., 2022), by adapting common mean process (Leroy et al., 2022), and by introducing neural embeddings or efficient frameworks (Liu et al., 2022; Chung & Al Kontar, 2023). However, the predictive performance of the LMC-based models may be inferior to that of isolated GPs when sufficient training data exists for each output (Bruinsma et al., 2020). See the results in Section 5.

---

[*]The corresponding author is Feng Yin. Our code is available at https://github.com/Blspdianna/GMOGP.

In this paper, we propose a novel multi-output regression framework, termed graphical MOGP (GMOGP). Unlike the LMC, which interprets the output correlations by covariance based on a set of shared GPs, the GMOGP, built upon a probability graphical model, can directly learn the conditional dependence and imply graphical representations of the multiple outputs. Moreover, learning the kernel hyperparameters of the GMOGP can be formulated as a multi-objective optimization (MOO) problem with Pareto optimal solutions. Further extensions by non-linear transformations make the transformed GMOGP (TGMOGP) available to fit non-Gaussian data. Enhanced predictive accuracy and efficiency validate the efficacy of the GMOGP across various synthetic/real datasets.

## 2 BACKGROUND

This section concludes basic concepts and notations in Gaussian process regression (GPR), serving as the bedrock of the upcoming GMOGP framework detailed in Section 3.

### 2.1 GAUSSIAN PROCESS REGRESSION

A GP characterizes a distribution over functions fully by a mean function $m(\mathbf{x})$ and a kernel function $k(\mathbf{x}, \mathbf{x}'; \boldsymbol{\theta})$, i.e.,

$$f(\mathbf{x}) \sim \mathcal{GP}\left(m(\mathbf{x}), k(\mathbf{x}, \mathbf{x}'; \boldsymbol{\theta})\right). \tag{1}$$

The commonly used squared exponential (SE) kernel, $k_{\text{SE}}(\mathbf{x}, \mathbf{x}'; l, \sigma_f^2) = \sigma_f^2 \cdot \exp(-\|\mathbf{x} - \mathbf{x}'\|^2 / l^2)$, is parametrized by a length-scale $l$ and a signal variance $\sigma_f^2$ (Rasmussen & Williams, 2006). Given a dataset $\mathcal{D} : \{X, \mathbf{y}\} = \{\mathbf{x}_n, y_n\}_{n=1}^N$, the GPR model can be described as:

$$y_n = f(\mathbf{x}_n) + \epsilon_n, \ n = 1, 2, \ldots, N, \tag{2}$$

where $\mathbf{x}_n \in \mathbb{R}^d$ denotes an input location, and the noise $\epsilon_n$ is assumed i.i.d with $\epsilon_n \sim \mathcal{N}(0, \sigma^2)$. In particular, the scalar-valued observations $\{y_n \in \mathbb{R}\}_{n=1}^N$ direct single-output GP (SOGP) models.

#### 2.1.1 MULTIOUTPUT GAUSSIAN PROCESS

For multi-output regression, an MOGP can be derived as:

$$\mathbf{f}(\mathbf{x}) \sim \mathcal{GP}(\boldsymbol{m}_M(\mathbf{x}), K_M(\mathbf{x}, \mathbf{x}'; \boldsymbol{\theta}_M)), \tag{3}$$

where the vector-valued function $\mathbf{f}(\mathbf{x}) = [f^{(1)}(\mathbf{x}), f^{(2)}(\mathbf{x}), \ldots, f^{(S)}(\mathbf{x})]_{(S>1)}$ is determined by the mean function $\boldsymbol{m}_M(\mathbf{x}) \in \mathbb{R}^S$ and the matrix-valued kernel function $K_M(\mathbf{x}, \mathbf{x}') \in \mathbb{R}^{S \times S}$. Note that the hyperparameters $\boldsymbol{\theta}_M$ are searched in a vector-valued reproducing kernel Hilbert space (RKHS).

At finite locations $X \in \mathbb{R}^{N \times d}$, the prior and the likelihood of the MOGP has the following forms:

$$(\text{Prior}): \qquad p(\mathbf{f}(X)) = \mathcal{N}(\boldsymbol{m}_M(X), K_M(X, X; \boldsymbol{\theta}_M)) \tag{4}$$

$$(\text{Likelihood}): \qquad p(Y|\mathbf{f}, X, \Sigma) = \mathcal{N}(\mathbf{f}(X), \Sigma \otimes I_N), \tag{5}$$

where $Y = [\mathbf{y}^{(1)}, \ldots \mathbf{y}^{(i)}, \ldots, \mathbf{y}^{(S)}] \in \mathbb{R}^{NS}$, and $\mathbf{y}^{(i)} = \{y_n^{(i)} \in \mathbb{R}\}_{n=1}^N$ are the labels of the $i^{th}$ output. The noise terms are assumed to have zero mean and a covariance matrix $\Sigma = \text{diag}\{\sigma_1^2, \sigma_2^2, \ldots, \sigma_S^2\}$. The kernel function for the MOGP prior evaluated at $X$ is derived as follows:

$$K_M(X, X; \boldsymbol{\theta}_M) = \begin{bmatrix} K_{11}(X, X; \boldsymbol{\theta}_{11}) & \cdots & K_{1S}(X, X; \boldsymbol{\theta}_{1S}) \\ \vdots & \ddots & \vdots \\ K_{S1}(X, X; \boldsymbol{\theta}_{S1}) & \cdots & K_{SS}(X, X; \boldsymbol{\theta}_{SS}) \end{bmatrix} \in \mathbb{R}^{SN \times SN}. \tag{6}$$

Generally, we train the MOGP models by minimizing their negative log marginal likelihood (NLL):

$$\mathcal{L}_{\{\boldsymbol{\theta}_M, \Sigma\}} \propto \tilde{Y}^T (K_M(X, X; \boldsymbol{\theta}_M) + \boldsymbol{\Sigma})^{-1} \tilde{Y} + \log |K_M(X, X; \boldsymbol{\theta}_M) + \boldsymbol{\Sigma}|, \tag{7}$$

with $\tilde{Y} = Y - \boldsymbol{m}_M(X)$, and $\boldsymbol{\Sigma} = \Sigma \otimes I_N$. Integrating over the parameter space differentiates the marginal likelihood methods from other non-Bayesian counterparts. The automatic incorporation of a trade-off between model fit and model complexity renders the marginal likelihood a valuable metric for model selection (Schölkopf et al., 2002; Micchelli & Pontil, 2005). Conditioning the joint Gaussian prior on the observations, the predictive distribution for a test input $\mathbf{x}_*$ turns out to be:

$$p(\mathbf{f}_*|\mathbf{x}_*, X, Y, \boldsymbol{\theta}_M) = \mathcal{N}(\bar{\mathbf{f}}_*, \mathbb{V}_*) \tag{8}$$

with (omitting kernel hyperparameters)

$$\begin{cases} \bar{\mathbf{f}}_* = K_M(\mathbf{x}_*, X)(K_M(X, X) + \boldsymbol{\Sigma})^{-1} Y & (9) \\ \mathbb{V}_* = K_M(\mathbf{x}_*, \mathbf{x}_*) - K_M(\mathbf{x}_*, X)(K_M(X, X) + \boldsymbol{\Sigma})^{-1} K_M(X, \mathbf{x}_*). & (10) \end{cases}$$

In order to infer the underlying functions and correlate the multiple outputs, the LMC models tailor distinct coefficients to each output via $Q$ shared independent GPs. The elements of their kernel functions (Eq.(6)) are formulated as $(K_{i,i'}(\mathbf{x}, \mathbf{x}')) = \sum_{q=1}^{Q} a_{iq} a_{i'q} k_q(\mathbf{x}, \mathbf{x}')$, where $i, i' \in \mathcal{I}, \mathcal{I} := \{1, 2, \ldots, S\}$, and different modes of the coefficients $a_{iq} a_{i'q}$ correspond to varied LMC variants (Alvarez et al., 2012; Liu et al., 2018). Although the LMC can generate dependent processes and measure output correlation, their applications are narrowed by two significant impediments: (i) The computational and storage burdens associated with the $SN$-dimensional correlation matrix, and (ii) model mismatch/inflexibility when the underlying likelihood (Eq.(5)) deviates from Gaussian distributions. Given these challenges, an alternative framework that can leverage the advantages of MOGPs, generate flexible prior/predictive distributions, and freely accommodate various efficient learning methods is considered in Section 3.

## 3 GRAPHICAL MULTIOUTPUT GAUSSIAN PROCESS

Ubiquitously, there exists interplay among the multiple outputs, giving rise to varied research, e.g., graph construction, transfer learning, and multi-task learning (Qiao et al., 2018; Vandenhende et al., 2021). For instance, the dependence (denoted by arrows) between a target output (marked in yellow) and the other six outputs can be described by a graphical representation shown in Figure 1(a). In the multi-output regression, the strong bond between the trend of the target $CO_2$ and $Temperature$ has been well discovered (solid link), while the other bonds are vague (dashed link) and worth exploring.

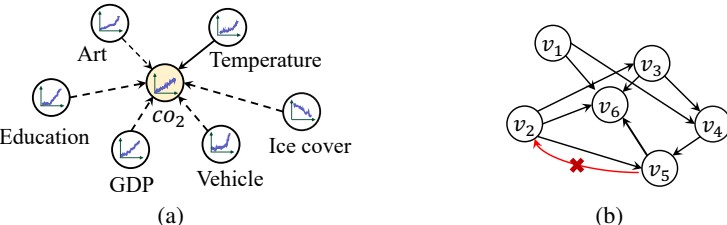

(a)          (b)

Figure 1: Illustrations of: (a) a graphical representation among the target $CO_2$ and the other outputs, (b) an example of a directed acyclic graph (the cross means that there are no directed cycles).

As directed graphical models (a.k.a. Bayesian network) can express the conditional dependence structure among random variables and make probabilistic statements for a broad class of distributions based on a specific graph (Koller & Friedman, 2009), we propose a graphical MOGP model to learn the output dependence jointly with the regression task.

Mathematically, a graph is built with nodes and links. In the directed graphical model, each node can represent random variables, while the links convey probabilistic dependence and have directionality indicated by arrows. For example, given nodes $\{v_1, v_2, \ldots, v_6\}$ and graph $\mathrm{G}_v$ shown in Figure 1(b), the joint distribution can be decomposed by repeatedly applying the product rule of probability:

$$p(v_1, v_2, \ldots, v_6) = p(v_6|v_1, v_2, v_3, v_4, v_5)p(v_5|v_1, v_2, v_3, v_4)\ldots p(v_2|v_1)p(v_1) \quad (11)$$

$$\overset{\mathrm{G}_v}{=} p(v_6|v_1, v_2, v_3, v_5)p(v_5|v_2, v_4)p(v_4|v_1, v_3)p(v_3|v_2)p(v_2)p(v_1). \quad (12)$$

The Eq.(12) is derived according to the conditional independence, e.g., $p(v_6|v_1, v_2, v_3, v_4, v_5) = p(v_6|v_1, v_2, v_3, v_5)$, as there is no link from node $v_4$ to $v_6$ in the graph $\mathrm{G}_v$. In more general terms, the joint distribution of multiple nodes $\boldsymbol{v} = \{v_i\}_{i \in \mathcal{I}}$ defined by a graph can be expressed as:

$$p(\boldsymbol{v}) = \prod_{i=1}^{S} p(v_i|\mathrm{pa}_i), \quad (13)$$

where $\mathrm{pa}_i$ denotes the parents[1] of $v_i$, e.g., $\mathrm{pa}_6 = \{v_1, v_2, v_3, v_5\}$ in Eq.(12). Note that the joint distribution only admits directed acyclic graphs; see more details in (Bishop & Nasrabadi, 2006).

---

[1]If there exists a link going from a node $i$ to a node $j$, the node $i$ is regarded as the parent of the node $j$.

Accordingly, for $S > 1$ outputs, we can model each output as an SOGP, and generate multivariate Gaussian random variables evaluated at $X$ (represented by node $f_X^{(j)}$, $j \in \mathcal{I}$). Following the result in Eq. (13), the joint distribution defined by the specific graph structure, with the target node $f_X^{(i)}$ (representing the $CO_2$) connected to the heads of arrows (see Figure 1(a)), can be derived as follows:

$$p(f_X^{(1)}, f_X^{(2)}, \ldots, f_X^{(S)}) = p(f_X^{(i)} | \mathrm{pa}_i) \prod_{j \in \mathrm{pa}_i} p(f_X^{(j)}). \tag{14}$$

Since the links are unknown in practice, the parents of the target node are assumed to contain all other nodes (except the target one) at first, i.e., $\mathrm{pa}_i = \{f_X^{(1)}, f_X^{(2)}, \ldots, f_X^{(S)}\} / \{f_X^{(i)}\}$, and can be adjusted via an attention mechanism detailed in Section 3.2. Alternately, we can treat each output as the target node (represented by $f_X^{(i)}, i \in \mathcal{I}$), and learn the conditional dependence on the other nodes. The diagrammatic illustration of the output bonds and the graph with respect to different target at the first stage (all vague) is given in Figure 2. Moreover, conditioning on the target node, the parents are dependent (see Remark 3.1), which paves the way to generate dependent processes.

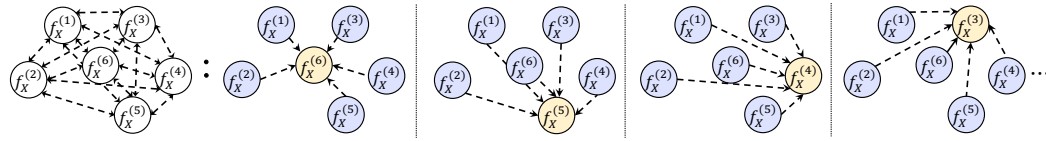

Figure 2: Illustration of the bonds among six outputs and the dependence structure between the different target (marked in yellow) and its parents (marked in purple) at the first stage.

**Remark 3.1** (Conditional Dependence). *Defined by the specific graph with joint distribution form:* $p(f_X^{(i)}, f_X^{(j)}, f_X^{(k)}) = p(f_X^{(i)} | f_X^{(j)}, f_X^{(k)}) p(f_X^{(j)}) p(f_X^{(k)})$, *if conditioning on the target $f_X^{(i)}$, the conditional distribution* $p(f_X^{(j)}, f_X^{(k)} | f_X^{(i)}) = p(f_X^{(j)}) p(f_X^{(k)}) p(f_X^{(i)} | f_X^{(j)}, f_X^{(k)}) / p(f_X^{(i)})$, *yielding the parents $f_X^{(j)}$ and $f_X^{(k)}$ are dependent, i.e., $f_X^{(j)} \not\perp f_X^{(k)} | f_X^{(i)}$.*

### 3.1 GRAPHICAL MULTIOUTPUT GAUSSIAN PROCESS PRIOR

Bayesian methods are invaluable within the machine learning community due to the ability to incorporate prior knowledge. Appropriate prior beliefs can represent detailed inductive bias, improve interpretability, and prevent overfitting (Theodoridis, 2015; Lotfi et al., 2022). In the GMOGP, we model the conditional distribution in Eq.(14) as Gaussian with aggregated information, namely,

$$p(f_X^{(i)} | \mathrm{pa}_i) = \mathcal{N}\Big( f_X^{(i)} \Big| \sum_{j \in \mathrm{pa}_i} \alpha_{i,j} f_X^{(j)} + \boldsymbol{m}_i, k_{\boldsymbol{\theta}_i}(X, X) \Big), \quad i \in \mathcal{I}, \tag{15}$$

where $\alpha_{i,j} \in \mathbb{R}$ and $\boldsymbol{m}_i$ are bias parameters added into the mean, and $k_{\boldsymbol{\theta}_i}(X, X) \in \mathbb{R}^{N \times N}$ is a covariance function. Conditioning on the states of its parents, each target node is of the form:

$$f_X^{(i)} = \sum_{j \in \mathrm{pa}_i} \alpha_{i,j} f_X^{(j)} + \boldsymbol{m}_i + \boldsymbol{\psi}_i, \tag{16}$$

with $\boldsymbol{m}_i$ and $\boldsymbol{\psi}_i \sim \mathcal{N}(\boldsymbol{0}, k_{\boldsymbol{\theta}_i}(X, X))$ characterizing the $i^{th}$ output. Given the parents with $p(f_X^{(j)}) = \mathcal{N}(\boldsymbol{m}_j, k_{\boldsymbol{\theta}_j}(X, X))$, we can derive the GMOGP prior via $\mathbb{E}(f_X^{(i)}) = \sum_{j \in \mathrm{pa}_i} \alpha_{i,j} \mathbb{E}(f_X^{(j)}) + \boldsymbol{m}_i$ and $\mathrm{cov}(f_X^{(i)}, f_X^{(i)}) = \mathbb{E}\Big[ (f_X^{(i)} - \mathbb{E}[f_X^{(i)}]) \Big\{ \sum_{j \in \mathrm{pa}_i} \alpha_{i,j} (f_X^{(j)} - \mathbb{E}[f_X^{(j)}]) + \boldsymbol{\psi}_i \Big\} \Big]$ (see Proposition 3.1).

**Proposition 3.1** (GMOGP Prior). *In the multi-output regression, defined by the specific graph with a target node and given its parents, the GMOGP prior for each target node ($i \in \mathcal{I}$) follows $p(f_X^{(i)}) = \mathcal{N}\Big( \sum_{j \in pa_i} \alpha_{i,j} \boldsymbol{m}_j + \boldsymbol{m}_i, \sum_{j \in pa_i} \alpha_{i,j}^2 k_{\boldsymbol{\theta}_j}(X, X) + k_{\boldsymbol{\theta}_i}(X, X) \Big)$. The proof is given in Appendix A.1.*

Following the noise setting in Section 2.1.1, the posterior of the $i^{th}$ output is proportional to the product of the prior and the likelihood, with hyperparameters $\Theta := \{\boldsymbol{\theta}_j\}_{j \in \mathcal{I}}$ and $\boldsymbol{\alpha}_i := \{\alpha_{i,j}\}_{j \in \mathrm{pa}_i}$,

$$p(\mathbf{y}^{(i)}, f_X^{(i)}; \Theta, \boldsymbol{\alpha}_i, \sigma_i) = p(f_X^{(i)}; \Theta, \boldsymbol{\alpha}_i) p(\mathbf{y}^{(i)} | f_X^{(i)}; \sigma_i). \tag{17}$$

Enhanced representation ability provided by the graph structure in the input space has been demonstrated in recent graph GP methods, such as Ng et al. (2018) and Fang et al. (2021), in which the graph structures among the input features are well discovered. For the multiple outputs, the graph structure is unclear and intriguing. To learn the graph while improving the performance of the multi-output regression, we propose to incorporate the attention mechanism into the GMOGP to modify the parents, as the attention-based graph construction methods have shown impressive performance, such as (Vaswani et al., 2017; Veličković et al., 2018; Kim & Oh, 2022).

## 3.2 Learning Parents with Attention

Centering on the $CO_2$ level (Figure 1(a)), the dependence on the number of vehicles, GDP level, and the degree of education is obscured. Intuitively, highly conditional dependent parents deserve large coefficients $\{\alpha_{i,j}\}_{j \in \text{pa}_i}$. Without knowing the graph adjacency, such coefficients can be learned by using an attention mechanism $\alpha_{i,j} = \exp(e_{i,j})/(1 + \sum_{j \in \text{pa}_i} \exp(e_{i,j}))$ and a scoring function:

$$e_{i,j} = \text{LeakyReLU}\left( \langle f_X^{(i)}, f_X^{(j)} \rangle \text{w}_{ij} + c_{ij} \right), \tag{18}$$

where the weights $\text{w}_{ij}$ and bias $c_{ij}$ are learning parameters. The inner product-based scoring function is widely used, as seen in works such as Vaswani et al. (2017) and Devlin et al. (2018), and can reveal geometric constructions involving angles, lengths, and distances (Schölkopf et al., 2002). An alternative scoring function employs the concatenation of inputs (Veličković et al., 2018; Brody et al., 2022). However, existing scoring functions parametrized by common parameter matrices showed static attention – a fixed node consistently achieves the highest attention (Brody et al., 2022).

In Eq.(18), we introduce a modified scoring function, which can achieve dynamic attention with the aid of disparate learning parameters and non-linear function. The learned coefficients of various real datasets are listed in Section 5 and Appendix B. In practice, we substitute the observation values $\langle \mathbf{y}^{(i)}, \mathbf{y}^{(j)} \rangle$ at the initial learning phase. When $\alpha_{i,j} \approx 0$, the parents set $\text{pa}_i$ is adjusted by unlinking the node $j$ ($j \in \mathcal{I}, j \neq i$). For classic MOGPs, the dependence between outputs is measured by symmetric covariance matrices. In comparison, our GMOGP can not only learn a more flexible asymmetric dependence measure but also capture the covariance from $\text{cov}(f_X^{(j)}, f_X^{(i)}) = \sum_{j' \in \text{pa}_i} \alpha_{i,j'} \text{cov}(f_X^{(j)}, f_X^{(j')}) + I_{ij} k_{\boldsymbol{\theta}_i}(X, X)$, where $I_{ij}$ is the $i, j$ element of the identity matrix (see Appendix A.1). In addition, the GMOGP can handle heterotopic data ($X^{(1)} \neq X^{(2)} \neq, \dots, \neq X^{(S)} \neq X$) by introducing an extra weight matrix to Eq.(18) for aligning extracted input features.

## 3.3 Model Learning and Inference

In Section 2.1.1, all kernel hyperparameters are learned through the marginal likelihood $\mathcal{L}_{\{\boldsymbol{\theta}_M, \Sigma\}}$. Both computational/storage requirements and learning challenges, brought about by the high-dimensional covariance matrix and multiple vector-valued RKHSs, impede the practical implementation of classic MOGPs (Yang et al., 2020). For the GMOGP model, each target node $i, i \in \mathcal{I}$ has its own prior and likelihood (Eq.(17)). Accordingly, the type II maximum likelihood can be conducted for every output, and the parameters corresponding to the $i^{th}$ output, i.e., $\boldsymbol{\gamma}^{(i)} := \{\Theta, \boldsymbol{\alpha}_i, \sigma_i\}$, can be updated via minimizing:

$$\mathcal{L}_{\boldsymbol{\gamma}^{(i)}}^{(i)} \propto \left\{ (\tilde{\mathbf{y}}^{(i)})^T \left( k_G^{(i)}(X, X) + \sigma_i^2 I_N \right)^{-1} \tilde{\mathbf{y}}^{(i)} + \log \left| k_G^{(i)}(X, X) + \sigma_i^2 I_N \right| \right\}, \tag{19}$$

where $\tilde{\mathbf{y}}^{(i)} = \mathbf{y}^{(i)} - (\sum_{j \in \text{pa}_i} \alpha_{i,j} \boldsymbol{m}_j + \boldsymbol{m}_i)$, and $k_G^{(i)}(X, X) = \sum_{j \in \text{pa}_i} \alpha_{i,j}^2 k_{\boldsymbol{\theta}_j}(X, X) + k_{\boldsymbol{\theta}_i}(X, X)$. Note that the size of the correlation matrix $k_G^{(i)}(X, X)$ is much smaller than the $K_M(X, X)$ in Eq.(6), and the hyperparameters $\Theta$ are shared among the multiple outputs for information exchange. Instead of solving with a high-dimensional problem, the separate objective with the shared kernel hyperparameters can be modeled by a multi-objective optimization (MOO) problem, i.e., $\boldsymbol{F}(\Theta) = [\mathcal{L}^{(1)}(\Theta), \mathcal{L}^{(2)}(\Theta), \dots, \mathcal{L}^{(S)}(\Theta)]^T$. Moreover, applying the weighted sum method with the objective function: $\sum_{i=1}^{S} w_i \mathcal{L}^{(i)}(\Theta)$, $w_i > 0$, to solve the MOO problem provides a sufficient condition for Pareto optimality[2] of the kernel hyperparameters (Marler & Arora, 2010). In the GMOGP, the

---

[2] A solution point is Pareto optimal if it is not possible to move from that point and improve at least one objective function without causing detriment to any other objective function (see details in Appendix A.5).

weights can be set equal, since there is no priority among the outputs, and each marginal likelihood $\mathcal{L}_{\boldsymbol{\gamma}^{(i)}}^{(i)}, i \in \mathcal{I}$ follows a Gaussian that can be normalized so that no one dominates the objective.

To summarize, the proposed GMOGP empowers the vanilla MOGPs with: (i) The attention-based asymmetric dependence measure ($\alpha_{ij} \neq \alpha_{ji}$), (ii) less computational and storage requirements, and (iii) Pareto optimal solutions of the kernel hyperparameters and extra graphical representations. Figure 3 presents the detailed workflow of the GMOGP hyperparameter optimization scheme. As a separable model, each target node possesses distinct training samples and parents, undergoing an independent inference procedure ($\mathcal{O}(N^3)$). It contributes individual information via $\boldsymbol{\theta}_i$ and aggregates dependent information through shared $\boldsymbol{\theta}_{j,\ j \in \mathrm{pa}_i}$, prompting the application of diverse distributed learning schemes. Key distinctions between the distributed GMOGP and the classic distributed GP (DGP) (Deisenroth & Ng, 2015) are summarized in Remark 3.2.

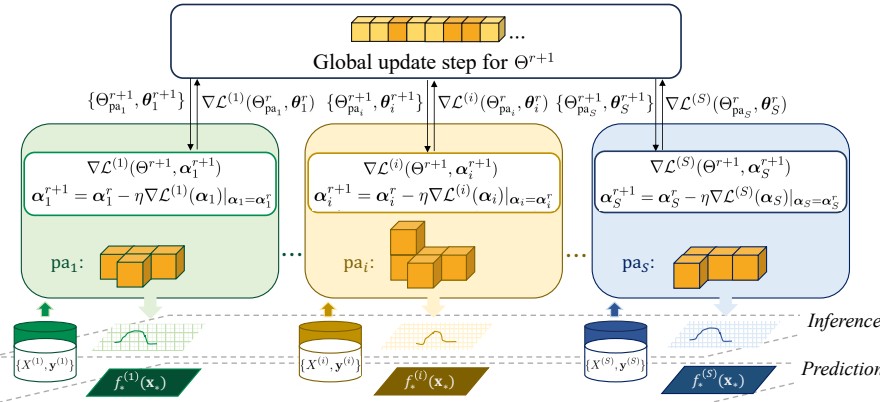

Figure 3: Illustrating the GMOGP workflow, each target output transmits individual knowledge while aggregating optimized information from its parents, enabling separable training and inference.

**Remark 3.2** (Distributed scheme). *The distributed framework can learn all parameters jointly and does not require adjustments, such as (i) simplifying approximations, e.g., the block diagonal approximation commonly employed in DGP approaches; (ii) extra prediction fusion methods, as the prediction at $\mathbf{x}_*$ can be separately inferred from $k_G^{(i)}(\mathbf{x}_*, X)(k_G^{(i)}(X, X) + \sigma_i^2 I_N)^{-1}\mathbf{y}^{(i)}$; (iii) data segmentation and (iv) additional client selection methods, as we can apply the identified parents.*

### 3.4 NON-GAUSSIAN GMOGP PRIOR

The Gaussian priors on functions are widely used in Bayesian machine learning literature, benefiting from posterior consistency, tractable inference, and closed-form solutions (Rasmussen & Williams, 2006). Whereas, the case of non-Gaussian prior/likelihood is often overlooked and ubiquitous in reality, e.g., heavy-tailed, bounded, and multimodal data (Snelson et al., 2003; Sendera et al., 2021).

Typical remedies include deep GP (DGP) (Damianou & Lawrence, 2013) and transformed GP (TGP) (Rios & Tobar, 2019). The performance of the TGP, transformed by marginal flows, was tested in (Maroñas et al., 2021) and demonstrated competitiveness with the DGP while incurring lower computational costs. However, the naive application of marginal flows to MOGP models involves untraceable learning procedures and inefficient high-dimensional approximation approaches, as 1D quadrature cannot be utilized. Lately, a TGP model, tailored for multi-class classification, has been proposed. It can generate dependent processes with different marginal flows from a single sample of a latent process (Maroñas & Hernández-Lobato, 2023). In the context of the multi-output regression problem addressed in this paper, the single latent function is neither sufficiently flexible nor accommodating distinct prior beliefs. Also, the strong dependence imposed by the single GP sample can degrade regression performance, especially when dealing with low-quality or biased outputs.

In contrast, the GMOGP model is unconstrained in applying non-linear transformations (termed flow). Concretely, extra model flexibility can be extended by compositing $K$ element-wise invertible transformations $\{\mathbb{G}_{\phi_k} : \mathcal{F} \mapsto \mathcal{F}\}_{k=0}^{K-1}$. Correspondingly, each target node defined in Eq.(16) aggregates non-linear relations, i.e., $\mathbb{G}_{\phi_k^{(i)}}(f_X^{(i)}) = \mathbb{G}_{\phi_k^{(i)}}\left(\sum_{j \in \mathrm{pa}_i} \alpha_{i,j} f_X^{(j)} + \boldsymbol{m}_i + \boldsymbol{\psi}_i\right)$. Applying

the inverse function theorem and the change of variable formula iteratively (Rezende & Mohamed, 2015), we can derive the transformed GMOGP prior as:

$$p_{\boldsymbol{\gamma}^{(i)},\Phi_i}\left(f_{K_X}^{(i)}|\mathbb{G},X\right) = p_{\boldsymbol{\gamma}^{(i)}}\left(f_{0_X}^{(i)}\right) \prod_{k=0}^{K-1} \left| \det \frac{\partial \mathbb{G}_{\phi_k^{(i)}}\left(f_{k_X}^{(i)}\right)}{\partial f_{k_X}^{(i)}} \right|^{-1} , \forall k \in \{0,1,\ldots,K-1\}, \quad (20)$$

where $f_{0_X}^{(i)} = f_X^{(i)}$, and $f_{k+1_X}^{(i)} = \mathbb{G}_{\phi_k^{(i)}}(f_{k_X}^{(i)})$ with parameters $\Phi_i = \{\phi_0^{(i)}, \phi_1^{(i)}, \ldots, \phi_{K-1}^{(i)}\}$, such as $\phi_k^{(i)} = \{\zeta_k, \rho_k, \lambda_k\}$ in Sinh-Archsinh (SAL) flow: $\mathbb{G}_{\phi_k^{(i)}}(x_k) = \zeta_k \cdot \sinh(\rho_k \cdot \operatorname{arcsinh}(x_k) - \lambda_k)$. We refer readers to Rios (2020) and Maroñas et al. (2021) for available flows and validity analysis.

Consolidating with the non-Gaussian prior, the transformed GMOGP (TGMOGP) can be easily implemented by adapting the off-the-shelf sparse variational inference algorithms in Hensman et al. (2015) or Maroñas et al. (2021) through optimizing the negative evidence lower bound (NELBO):

$$\min_{\substack{\{\boldsymbol{\gamma}^{(i)},\Phi_i,\boldsymbol{u}_0^{(i)},\boldsymbol{m}_u^{(i)},K_u^{(i)}\}; \\ i=1,2,\ldots,S}} -\left( \sum_{i=1}^{S} \mathbb{E}_{q\left(f_{0_X}^{(i)}\right)}\left[\log p\left(\mathbf{y}^{(i)}|\mathbb{G}_{\Phi_i}(f_{0_X}^{(i)})\right)\right] + \mathbb{E}_{q\left(\boldsymbol{u}_0^{(i)}\right)}\left[\log \frac{p(\boldsymbol{u}_0^{(i)})}{q(\boldsymbol{u}_0^{(i)})}\right] \right) \quad (21)$$

with inducing points $\boldsymbol{u}_0^{(i)} \in \mathbb{R}^M, M \ll N$ and $q(\boldsymbol{u}_0^{(i)}) = \mathcal{N}(\boldsymbol{m}_u^{(i)}, K_u^{(i)})$. The detailed derivation of the NELBO and variational gap are elaborated in Appendix A.2. For inference, by the law of the unconscious statistician (LOTUS), the predictions of output $i, i \in \mathcal{I}$ can be estimated from the first moment of predictive distribution: $p(\mathbf{y}_*^{(i)}) = \int p(\mathbf{y}_*^{(i)}|\mathbb{G}_{\Phi_i}(f_{0_X}^{(i)})) q(f_{0_X}^{(i)}) df_{0_X}^{(i)}$ (Snelson et al., 2003). In total, the dependencies of the variables in the GMOGP are concluded in the following Figure 4.

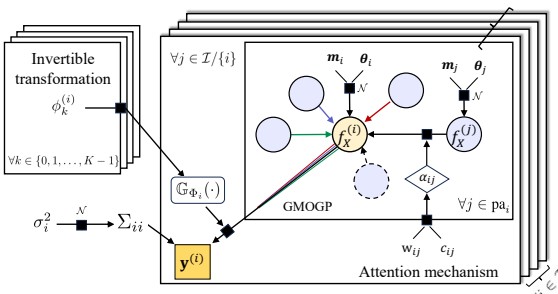

Figure 4: Visualizing variable dependencies in the GMOGP. Each target output has their own parents, transformations, and samples with knowledge exchanged by kernel hyperparameters $\boldsymbol{\theta}_{j,\ j \in \text{pa}_i}$.

## 4   RELATED WORK

The graph-based GP models have been widely developed in local structured feature spaces and relational learning discipline, resulting in the development of graph GP (Ng et al., 2018) and relational GP (Chu et al., 2006; Silva et al., 2007). Without knowing neighbor graph structures or a complete Laplacian matrix, attention-based graph construction methods have sprung up and have been shown to be successful in supervised learning problems (Veličković et al., 2018). The idea of finding neighborhoods using non-negative kernel (NNK) regression (Shekkizhar & Ortega, 2020) explores geometric understanding and connections to the classic matching pursuit approach (Tropp & Gilbert, 2007). In the GMOGP framework, we can draw parallels with the kernel matching pursuit objective (Vincent & Bengio, 2002), where $\mathbf{y}_*^{(i)} = \tilde{\boldsymbol{\beta}} k_G^{(i)}(X_*, X)$ can be regarded as matching through constructed kernel dictionary and coefficient $\tilde{\boldsymbol{\beta}} = (k_G^{(i)}(X,X) + \sigma_i^2 I_N)^{-1} \mathbf{y}^{(i)}$. Moreover, attention mechanisms have been utilized in the multi-task learning community, such as the multi-task attention networks (MTAN) in (Liu et al., 2019). In addition, the weighted sum objective function has been well-established for balancing task contributions (Vandenhende et al., 2021).

Regarding to the graph among the multiple outputs, the GMOGP introduces the probability graphical model to the MOGP regression, leading to an efficient MOGP alternative with attention-based flexible/non-Gaussian priors and various graphical representations.

Table 1: The average test RMSE of the synthetic experiments described in Section 5.1. All metrics are compared against the baselines: [1] Isolated SOGPs, [2] LMC (de Wolff et al., 2021), [3] free-form task similarity model (FICM) (Bonilla et al., 2007), [4] Gaussian process regression network (GPRN) (Wilson et al., 2012), and [5] convolution process (CMOGP) (Alvarez & Lawrence, 2011). ($l_{NF}$ : The flow parameters, $V_m$ : The variational parameters.)

| | Average RMSE | Test NLL | $K_{\text{dim}}$ | Number of Parameters |
|---|---|---|---|---|
| [1] SOGP | $0.5653\pm0.0023$ | $0.4891\pm0.0043$ | $N$ | $4S$ |
| [2] LMC | $0.5917\pm0.0096$ | $0.5543\pm0.0506$ | $S \times N$ | $(2+S)Q + 2S + 1$ |
| [3] FICM | $0.5544\pm0.0046$ | $0.4798\pm0.0176$ | $S \times N$ | $(S(S+5)+4)/2$ |
| [4] GPRN | $0.5819\pm0.0207$ | $0.5787\pm0.0445$ | $S \times N$ | $2(S+Q) + 3$ |
| [5] CMOGP | $0.5539\pm0.0089$ | $0.4689\pm0.0143$ | $S \times N$ | $(2+S)Q + 2S + 1$ |
| [6] GMOGP | $0.5541\pm0.0054$ | $0.1636\pm0.0143$ | $N$ | $S(S+5)+1$ |
| [7] TGMOGP | $0.5343\pm0.0023$ | $-0.6354\pm0.0023$ | $N$ | $S(S+5)+1+3l_{NF}+V_m$ |

## 5 EXPERIMENTS

In this section, we benchmark the predictive performance of the proposed (transformed) GMOGP against baseline MOGP methods and isolated SOGP using various synthetic and real-world datasets.

### 5.1 SYNTHETIC DATA EXPERIMENTS

A multi-output regression task with non-Gaussian noise and different function compositions is evaluated. Five outputs ($S = 5$) are generated by the following functions specialized at $X \in \mathbb{R}^{1800\times2}$:

$$\mathbf{y}^{(1)} = f_1(X) + \boldsymbol{\epsilon}_1, \tag{22}$$
$$\mathbf{y}^{(2)} = f_1(X) + f_2(X) + \boldsymbol{\epsilon}_2, \tag{23}$$
$$\mathbf{y}^{(3)} = \sinh(2\operatorname{arcsinh}(f_1(X) + f_2(X)) + \boldsymbol{\epsilon}_3), \tag{24}$$
$$\mathbf{y}^{(4)} = 3\tanh(f_3(X)f_4(X) + f_1(X) + \boldsymbol{\epsilon}_4), \tag{25}$$
$$\mathbf{y}^{(5)} = 5f_3(X)f_4(X) + \boldsymbol{\epsilon}_5, \tag{26}$$

where $f_1(\mathbf{x}) = 2\cos(x_1+x_2)$, $f_2(\mathbf{x}) = (x_1+x_2)^2$, $f_3(\mathbf{x}) = \exp(|x_1 x_2|+1)$, $f_4(\mathbf{x}) = \log(x_1+3)$, and $\epsilon_1, \epsilon_2, \ldots, \epsilon_5$ are i.i.d. Gaussian noise with a common standard deviation 0.2. The detailed experiment settings and available marginal flows are elaborated in Appendix B.

Table 1 shows the compared predictive performance at 600 test points. The GMOGP is competitive with the FICM but exhibits a smaller test NLL, and surpasses other baselines in both test RMSE and NLL. Given the test error for each output shown in Figure 5(c), the TGMOGP achieves improved performance, especially in the $3^{rd}$ output, confirming the enhanced representation ability w.r.t. the non-Gaussian data. The inferior performance of the classic MOGP methods (even worse than the isolated SOGP) indicates inefficient model learning and correlation measures in high-dimensional space. Moreover, increasing the number of independent latent processes $Q$ in LMC variants yields little performance gain (shown in Figure 5(b)). Other results are obtained with $Q = S$. By contrast, our TGMOGP achieves superior results constantly among all different training sizes (Figure 5(a)).

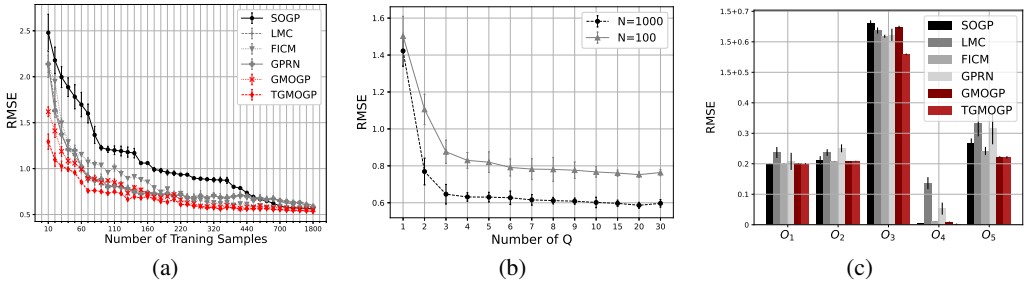

Figure 5: The sub-figures show (a) the average RMSE changes with the number of training samples, (b) the RMSE versus the number of latent independent GPs, and (c) the test error of each output.

Table 2: Comparison of test RMSE on real datasets, where SGPRN (Li et al., 2021) and variational LMC (V-LMC) are tested. The shadowed results are learned with two distributed computing units.

| Datasets | SOGP | V-LMC$_{100}$ | FICM | SGPRN | GMOGP | TGMOGP$_{100}$ |
|---|---|---|---|---|---|---|
| JURA | 0.605±0.01 | 0.443±0.01 | 0.394±0.05 | 0.438±0.02 | **0.376±0.01** | 0.382±0.01 |
| ECG | 0.245±0.01 | 0.229±0.01 | 0.222±0.01 | 0.232±0.02 | 0.219±0.00 | **0.217±0.00** |
| EEG | 0.343±0.05 | 0.207±0.03 | 0.147±0.03 | 0.261±0.03 | **0.082±0.01** | 0.117±0.00 |
| SARCOS$_1$ | 1.139±0.01 | 1.063±0.01 | 0.792±0.04 | 0.844±0.04 | 0.643±0.03 | **0.558±0.00** |
| KUKA | 0.05±0.01 | 0.14±0.01 | 0.03±0.01 | 0.12±0.02 | **0.02/ 0.02 ±0.00** | 0.04/ 0.04 ±0.01 |
| Test NLL | -0.25±0.01 | -0.51±0.01 | -0.65 ±0.02 | -0.55±0.01 | -1.81/ -1.76 ±0.02 | **-3.49/ -3.48 ±0.01** |
| SARCOS$_2$ | 0.26±0.05 | 0.29±0.04 | 0.33±0.03 | - | 0.21/ 0.22 ±0.02 | **0.16/ 0.16 ±0.01** |
| Time/Iter | 20.75(s) | 370.2(s) | 419.3(s) | >2400(s) | 32.41/ 21.07 (s) | **4.65/ 3.43 (s)** |

Table 3: Instances of the learned attention coefficients values from the GMOGP ($\alpha_{jj} = 1$).

| Coefficient | $\alpha_{1,2}$ | $\alpha_{1,3}$ | $\alpha_{1,4}$ | $\alpha_{2,1}$ | $\alpha_{2,3}$ | $\alpha_{2,4}$ | $\alpha_{3,1}$ | $\alpha_{3,2}$ | $\alpha_{3,4}$ | $\alpha_{4,1}$ | $\alpha_{4,2}$ | $\alpha_{4,3}$ |
|---|---|---|---|---|---|---|---|---|---|---|---|---|
| SARCOS$_2$ | 1.9e-4 | 0.998 | 2.9e-4 | 5.4e-4 | 6.9e-4 | 0.898 | 7.3e-5 | 5.1e-5 | 0.989 | 1.3e-3 | 0.966 | 6.7e-4 |

## 5.2 REAL-WORLD DATA EXPERIMENTS

In this section, we evaluate the predictive performance of the GMOGP alongside scalable MOGP models in real-world applications. The data description and tasks are detailed in Appendix B.3, where we select the first 2k/20k training samples (denoted as SARCOS$_{1/2}$) from the SARCOS dataset to investigate how performance varies with different training sizes. The results in Table 2 show that our GMOGP outperforms baseline models on all real datasets in accuracy and efficiency. For variational inference with 100 inducing points, the TGMOGP$_{100}$ achieves lower predictive error than the V-LMC$_{100}$ with largely reduced time cost, and even surpasses the models trained with full data. One probable reason is that the variational distribution generated from a free-form Gaussian in the TGMOGP$_{100}$ is also transformed by the SAL flow, which improves the possibility of learning a sufficient statistic. Another reason is that the TGMOGP transformed by the SAL flow can describe asymmetry and non-Gaussian tailed distributions (Jones & Pewsey, 2019), and the SARCOS data is negatively-skewed with mean $<$ median $<$ mode. The trend illustrated in the Figure 5(a) is also shown by the different rankings across the two SARCOS data. As for the two larger datasets (KUKA (12k) and SARCOS$_2$), we learn the GMOGP-based models through the distributed learning framework using two distributed computing units (the shadowed results), where comparable results are achieved with reduced time consumption. In a comparison of the test negative log-likelihood, the outstanding result of the TGMOGP model implies it has a more appropriate model complexity.

The graphical representations learned along with the improved predictive performance on the multi-output regression tasks (KUKA and SARCOS$_2$) can be indicated by the attention coefficients (Table 3). In the SARCOS (4-outputs), we can identify the parent of the node representing the $2^{nd}$ output is the node corresponding to the $4^{th}$ output ($\alpha_{2,4} = 0.8978$), since we unlink the other two nodes with coefficients approaching 0. Similarly, the dependence between the $3^{rd}$ and $4^{th}$ outputs can be implied by $\alpha_{3,4} = 0.9998$. To understand the graphical representations, we calculate the Pearson correlation coefficients between the $2^{nd}$/$3^{rd}$ and $4^{th}$ outputs, which give high values of 0.7744 and 0.9657 coincidently. Other interpretations of a traffic prediction task are explored in Appendix B.3.

## 6 CONCLUSION

We propose a graphical multioutput Gaussian process (GMOGP) model, an innovative framework tailored for efficient multi-output regression and graphical representation learning. Noteworthy model flexibility and optimality are verified by superior predictive performance achieved across various synthetic and real datasets. Distributed learning frameworks and sparse variational inference methods can be directly applied to the proposed GMOGP framework, giving a chance to deal with large datasets and a great number of outputs. Moreover, varied graphical representations and conditional dependence empower the vanilla MOGP models with more representation ability.

## 7 ACKNOWLEDGMENTS

This work was supported by the NSFC under Grant No. 62271433, and in part by Shenzhen Science and Technology Program under Grant No. JCYJ20220530143806016 and No. RCJC20210609104448114.

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

# A MATHEMATICAL APPENDIX

In the part A, we will give some insights and detailed derivation of the graphical multioutput Gaussian process (GMOGP) prior, the evidence lower bound (ELBO) of the transformed GMOGP, etc.

## A.1 GMOGP PRIOR DERIVATION

In the main paper, we can derive the target node $i$ in the GMOGP as the following aggregated form, i.e.,

$$f_X^{(i)} = \sum_{j \in \mathrm{pa}_i} \alpha_{i,j} f_X^{(j)} + \boldsymbol{m}_i + \boldsymbol{\psi}_i. \tag{27}$$

Recalling that each node represents the random variables generated by a single output Gaussian process, i.e., we have $p(f_X^{(j)}) = \mathcal{N}(\boldsymbol{m}_j, k_{\boldsymbol{\theta}_j}(X, X)), j \in \mathrm{pa}_i$ and $\boldsymbol{\psi}_i \sim \mathcal{N}(\boldsymbol{0}, k_{\boldsymbol{\theta}_i}(X, X))$. Therefore, the GMOGP prior of target node $p(f_X^{(i)})$ follows Gaussian with mean:

$$\mathbb{E}(f_X^{(i)}) = \sum_{j \in \mathrm{pa}_i} \alpha_{i,j} \mathbb{E}(f_X^{(j)}) + \boldsymbol{m}_i \tag{28}$$

$$= \sum_{j \in \mathrm{pa}_i} \alpha_{i,j} \boldsymbol{m}_j + \boldsymbol{m}_i \tag{29}$$

and covariance:

$$\mathrm{cov}(f_X^{(i)}, f_X^{(i)}) = \mathbb{E}\left[\left(f_X^{(i)} - \mathbb{E}[f_X^{(i)}]\right)\left(f_X^{(i)} - \mathbb{E}[f_X^{(i)}]\right)^T\right] \tag{30}$$

$$= \mathbb{E}\left[\left(f_X^{(i)} - \mathbb{E}[f_X^{(i)}]\right)\left\{\sum_{j \in \mathrm{pa}_i} \alpha_{i,j}\left(f_X^{(j)} - \mathbb{E}[f_X^{(j)}]\right) + \boldsymbol{\psi}_i\right\}\right] \tag{31}$$

$$= \sum_{j \in \mathrm{pa}_i} \alpha_{i,j} \mathrm{cov}(f_X^{(i)}, f_X^{(j)}) + k_{\boldsymbol{\theta}_i}(X, X), \tag{32}$$

where,

$$\mathrm{cov}(f_X^{(i)}, f_X^{(j)}) = \mathrm{cov}(f_X^{(j)}, f_X^{(i)}) \tag{33}$$

$$= \mathbb{E}\left[\left(f_X^{(j)} - \mathbb{E}[f_X^{(j)}]\right)\left(f_X^{(i)} - \mathbb{E}[f_X^{(i)}]\right)^T\right] \tag{34}$$

$$= \mathbb{E}\left[\left(f_X^{(j)} - \mathbb{E}[f_X^{(j)}]\right)\left\{\sum_{k \in \mathrm{pa}_i} \alpha_{i,k}\left(f_X^{(k)} - \mathbb{E}[f_X^{(k)}]\right) + \boldsymbol{\psi}_i\right\}\right] \tag{35}$$

$$= \sum_{k \in \mathrm{pa}_i} \alpha_{i,k} \mathrm{cov}(f_X^{(j)}, f_X^{(k)}) \quad (i \neq j). \tag{36}$$

Notably, under the specific graphical representation (such as the decomposition in Figure 2), there are no parents of other nodes $j$, i.e., $\mathrm{pa}_{j,j \neq i} = \{\emptyset\}$. Then, we have $\sum_{k \in \mathrm{pa}_i} \alpha_{i,k} \mathrm{cov}(f_X^{(j)}, f_X^{(k)}) = \alpha_{i,j} \mathrm{cov}(f_X^{(j)}, f_X^{(j)}) = \alpha_{i,j} k_{\boldsymbol{\theta}_j}(X, X)$. As a result, substituting this result to the above Eq.(32), we can derive the covariance form for our target node $i$, i.e.

$$k_G^{(i)}(X, X) := \mathrm{cov}(f_X^{(i)}, f_X^{(i)}) = \sum_{j \in \mathrm{pa}_i} \alpha_{i,j}^2 k_{\boldsymbol{\theta}_j}(X, X) + k_{\boldsymbol{\theta}_i}(X, X). \tag{37}$$

**Remark A.1.** *In the context of the probabilistic graphic model, only directed acyclic graphs (DAGs) can be generated and represented. Given a well-defined ordering w.r.t. the joint distribution (see Remark A.2), the covariance can be evaluated recursively starting from the lowest numbered node.*

**Remark A.2** (Ordering)**.** *The decomposed product term of the joint distribution is asymmetrical, i.e., an implicit order must be chosen, such as $(v_1, v_2, \ldots, v_6)$ in Eq.(12). Following a different ordering, we can obtain a different decomposition and hence a different graphical representation.*

## A.2 ELBO OF THE TRANSFORMED GMOGP

Following a similar derivation as (Titsias, 2009) and (Maroñas et al., 2021), the log marginal likelihood of output $i, i = 1, 2, \ldots, S$ admits the following inequality: (We denote specific locations $n$ with additional subscripts, such that $f_{K_{X_n}}^{(i)} = \mathbb{G}_{\mathbf{\Phi}_i}\left(f_{0_{X_n}}^{(i)}\right) = \mathbb{G}_{\phi_{K-1}^{(i)}} \circ \cdots \circ \mathbb{G}_{\phi_1^{(i)}} \circ \mathbb{G}_{\phi_0^{(i)}}\left(f_{0_{X_n}}^{(i)}\right)$, with $f_{0_X}^{(i)} = f_X^{(i)}$)

$$\log p(\mathbf{y}^{(i)}) \geq \mathbb{E}_{q\left(f_{K_X}^{(i)}, \mathbf{u}_K^{(i)}\right)} \left[ \log \frac{p\left(\mathbf{y}^{(i)} \mid f_{K_X}^{(i)}\right) p\left(f_{K_X}^{(i)}, \mathbf{u}_K^{(i)}\right)}{q\left(f_{K_X}^{(i)}, \mathbf{u}_K^{(i)}\right)} \right] \tag{38}$$

$$= \mathbb{E}_{q\left(f_{K_X}^{(i)}, \mathbf{u}_K^{(i)}\right)} \left[ \log \frac{\prod_n p\left(\mathbf{y}_n^{(i)} \mid f_{K_{X_n}}^{(i)}\right) p\left(f_{K_X}^{(i)}, \mathbf{u}_K^{(i)}\right)}{q\left(f_{K_X}^{(i)}, \mathbf{u}_K^{(i)}\right)} \right] \tag{39}$$

$$= \underbrace{\sum_n^N \mathbb{E}_{q\left(f_{K_X}^{(i)}, \mathbf{u}_K^{(i)}\right)} \left[ \log p\left(\mathbf{y}_n^{(i)} \mid f_{K_{X_n}}^{(i)}\right) \right]}_{\text{ELL}^{(i)}} + \underbrace{\mathbb{E}_{q\left(f_{K_X}^{(i)}, \mathbf{u}_K^{(i)}\right)} \left[ \log \frac{p\left(f_{K_X}^{(i)}, \mathbf{u}_K^{(i)}\right)}{q\left(f_{K_X}^{(i)}, \mathbf{u}_K^{(i)}\right)} \right]}_{-\text{KL}^{(i)}}, \tag{40}$$

where we define the approximate posterior as:

$$q(f_{K_X}^{(i)}, \mathbf{u}_K^{(i)}) = p(f_{K_X}^{(i)} | \mathbf{u}_K^{(i)}) q(\mathbf{u}_K^{(i)}) \tag{41}$$

$$= p(f_{K_X}^{(i)} | \mathbf{u}_K^{(i)}) q(\mathbf{u}_0^{(i)}) \prod_{k=0}^{K-1} \left| \det \frac{\partial \mathbb{G}_{\phi_k^{(i)}}(\mathbf{u}_k^{(i)})}{\partial \mathbf{u}_k^{(i)}} \right|^{-1}. \tag{42}$$

Here,

$$p(f_{K_X}^{(i)} | \mathbf{u}_K^{(i)}) = p(f_{K_X}^{(i)}, \mathbf{u}_K^{(i)}) / p(\mathbf{u}_K^{(i)}) \tag{43}$$

$$= p(f_{0_X}^{(i)}, \mathbf{u}_0^{(i)}) \prod_{k=0}^{K-1} \left| \det \begin{bmatrix} \frac{\partial \mathbb{G}_{\phi_k^{(i)}}(f_{k_X}^{(i)})}{\partial f_{k_X}^{(i)}} & \frac{\partial \mathbb{G}_{\phi_k^{(i)}}(f_{k_X}^{(i)})}{\partial \mathbf{u}_k^{(i)}} \\ \frac{\partial \mathbb{G}_{\phi_k^{(i)}}(\mathbf{u}_k^{(i)})}{\partial f_{k_X}^{(i)}} & \frac{\partial \mathbb{G}_{\phi_k^{(i)}}(\mathbf{u}_k^{(i)})}{\partial \mathbf{u}_k^{(i)}} \end{bmatrix} \right|^{-1} \Bigg/ p(\mathbf{u}_K^{(i)}) \tag{44}$$

$$= p(f_{0_X}^{(i)}, \mathbf{u}_0^{(i)}) \Bigg/ \left( p(\mathbf{u}_0^{(i)}) \prod_{k=0}^{K-1} \left| \det \frac{\partial \mathbb{G}_{\phi_k^{(i)}}(\mathbf{u}_k^{(i)})}{\partial \mathbf{u}_k^{(i)}} \right|^{-1} \right)$$

$$\times \prod_{k=0}^{K-1} \left| \det \left[ \frac{\partial \mathbb{G}_{\phi_k^{(i)}}(f_{k_X}^{(i)})}{\partial f_{k_X}^{(i)}} - \underbrace{\frac{\partial \mathbb{G}_{\phi_k^{(i)}}(f_{k_X}^{(i)})}{\partial \mathbf{u}_k^{(i)}} \left( \frac{\partial \mathbb{G}_{\phi_k^{(i)}}(\mathbf{u}_k^{(i)})}{\partial \mathbf{u}_k^{(i)}} \right)^{-1} \frac{\partial \mathbb{G}_{\phi_k^{(i)}}(\mathbf{u}_k^{(i)})}{\partial f_{k_X}^{(i)}}}_{\text{evaluated to zero for marginal flows}} \right] \det \frac{\partial \mathbb{G}_{\phi_k^{(i)}}(\mathbf{u}_k^{(i)})}{\partial \mathbf{u}_k^{(i)}} \right|^{-1} \tag{45}$$

$$= p(f_{0_X}^{(i)} | \mathbf{u}_0^{(i)}) \prod_{k=0}^{K-1} \left| \det \left[ \frac{\partial \mathbb{G}_{\phi_k^{(i)}}(f_{k_X}^{(i)})}{\partial f_{k_X}^{(i)}} \right] \right|^{-1}. \tag{46}$$

Moreover, applying marginal flows (coordinate-wise), we can derive a more simplified form:

$$\det \frac{\partial \mathbb{G}_{\phi_k^{(i)}}(f_{k_X}^{(i)})}{\partial f_{k_X}^{(i)}} = \prod_{n=1}^N \frac{d\mathbb{G}_{\phi_k^{(i)}}(f_{k_{X_n}}^{(i)})}{df_{k_{X_n}}^{(i)}}, \tag{47}$$

and we suppose

$$q(\boldsymbol{u}_0^{(i)}) = \mathcal{N}(\boldsymbol{m}_u^{(i)}, K_u^{(i)}) \tag{48}$$

with $\boldsymbol{m}_u^{(i)} \in \mathbb{R}^M$ and $K_u^{(i)} \in \mathbb{R}^{M \times M}$ being the variational parameters. The detailed derivation and simplification of the expected log-likelihood (ELL) and KL divergence in Eq.(40) are specified separately in the next subsections.

### A.2.1    KL DIVERGENCE FOR A TARGET OUTPUT

Following the approximate posterior in Eq.(42), the conditional term in the $\text{KL}^{(i)}$ can be canceled, namely:

$$\text{KL}^{(i)} = -\mathbb{E}_{q\left(f_{K_X}^{(i)}, \boldsymbol{u}_K^{(i)}\right)} \left[ \log \frac{p\left(f_{K_X}^{(i)}, \boldsymbol{u}_K^{(i)}\right)}{q\left(f_{K_X}^{(i)}, \boldsymbol{u}_K^{(i)}\right)} \right] \tag{49}$$

$$= -\mathbb{E}_{q\left(\boldsymbol{u}_K^{(i)}\right)} \left[ \log \frac{\cancel{p(f_{K_X}^{(i)}|\boldsymbol{u}_K^{(i)})} p(\boldsymbol{u}_0^{(i)}) \prod_{k=0}^{K-1} \left| \det \cancel{\frac{\partial \mathbb{G}_{\phi_k^{(i)}}(\boldsymbol{u}_k^{(i)})}{\partial \boldsymbol{u}_k^{(i)}}} \right|^{-1}}{\cancel{p(f_{K_X}^{(i)}|\boldsymbol{u}_K^{(i)})} q(\boldsymbol{u}_0^{(i)}) \prod_{k=0}^{K-1} \left| \det \cancel{\frac{\partial \mathbb{G}_{\phi_k^{(i)}}(\boldsymbol{u}_k^{(i)})}{\partial \boldsymbol{u}_k^{(i)}}} \right|^{-1}} \right] \tag{50}$$

$$= -\mathbb{E}_{q\left(\boldsymbol{u}_K^{(i)}\right)} \left[ \log \frac{p(\boldsymbol{u}_0^{(i)})}{q(\boldsymbol{u}_0^{(i)})} \right] \tag{51}$$

$$= -\mathbb{E}_{q\left(\boldsymbol{u}_0^{(i)}\right)} \left[ \log \frac{p(\boldsymbol{u}_0^{(i)})}{q(\boldsymbol{u}_0^{(i)})} \right], \quad \left( \iff \text{KL}\left[ q(\boldsymbol{u}_0^{(i)}) \,\|\, p(\boldsymbol{u}_0^{(i)}) \right] \right) \tag{52}$$

where we first marginalized $f_{K_X}^{(i)}$ and then applied the law of the unconscious statistician (LOTUS). Concretely, given an invertible transformation $\mathbb{G}(\cdot)$ (any transformation implies valid stochastic processes, we refer the readers to (Rios, 2020) for more details), the expectation of any function $g(\cdot)$ under the distribution $p(\boldsymbol{u}_K^{(i)})$ admits:

$$\mathbb{E}_{p(\boldsymbol{u}_K^{(i)})} \left[ g(\boldsymbol{u}_K^{(i)}) \right] = \mathbb{E}_{p(\boldsymbol{u}_0^{(i)})} \left[ g(\mathbb{G}_{\boldsymbol{\Phi}_i}(\boldsymbol{u}_0^{(i)})) \right]. \tag{53}$$

### A.2.2    EXPECTED LOG-LIKELIHOOD UNDER MARGINAL FLOWS

After integrating out the $\boldsymbol{u}_K^{(i)}$ and applying the LOTUS rule over the expectation with regard to $q(f_{K_X}^{(i)})$, we can simplify the $\text{ELL}^{(i)}$ term in Eq.(40) as:

$$\text{ELL}^{(i)} = \sum_n^N \mathbb{E}_{q\left(f_{K_X}^{(i)}, \boldsymbol{u}_K^{(i)}\right)} \left[ \log p\left(\mathbf{y}_n^{(i)} \mid f_{K_{X_n}}^{(i)}\right) \right] \tag{54}$$

$$= \sum_n^N \int \left[ \log p\left(\mathbf{y}_n^{(i)} \mid f_{K_{X_n}}^{(i)}\right) \right] q\left(f_{K_X}^{(i)} | \boldsymbol{u}_K^{(i)}\right) p\left(\boldsymbol{u}_K^{(i)}\right) d\boldsymbol{u}_K^{(i)} df_{K_X}^{(i)} \tag{55}$$

$$= \sum_n^N \mathbb{E}_{q\left(f_{K_X}^{(i)}\right)} \left[ \log p\left(\mathbf{y}_n^{(i)} \mid f_{K_{X_n}}^{(i)}\right) \right] \tag{56}$$

$$= \sum_n^N \mathbb{E}_{q\left(f_{0_X}^{(i)}\right)} \left[ \log p\left(\mathbf{y}_n^{(i)} \mid \mathbb{G}_{\boldsymbol{\Phi}_i}(f_{0_{X_n}}^{(i)})\right) \right]. \tag{57}$$

The distribution $q\big(f_{0_X}^{(i)}\big)$ can be calculated by:

$$q\big(f_{0_X}^{(i)}\big) = \int p(f_{0_X}^{(i)}|\boldsymbol{u}_0^{(i)})q(\boldsymbol{u}_0^{(i)})d\boldsymbol{u}_0^{(i)} \tag{58}$$

$$= \mathcal{N}\Big(k_G^{(i)}(X,Z)(k_G^{(i)}(Z,Z))^{-1}\boldsymbol{m}_u^{(i)},$$

$$k_G^{(i)}(X,X) - k_G^{(i)}(X,Z)(k_G^{(i)}(Z,Z))^{-1}\left[k_G^{(i)}(Z,Z) + K_u^{(i)}\right](k_G^{(i)}(Z,Z))^{-1}k_G^{(i)}(Z,X)\Big), \tag{59}$$

with

$$p(\boldsymbol{u}_0^{(i)}) = \mathcal{N}\left(\boldsymbol{0}, k_G^{(i)}(Z,Z)\right) \tag{60}$$

$$p(f_{0_X}^{(i)}|\boldsymbol{u}_0^{(i)}) = \mathcal{N}\Big(k_G^{(i)}(X,Z)(k_G^{(i)}(Z,Z))^{-1}\boldsymbol{u}_0^{(i)},$$

$$k_G^{(i)}(X,X) - k_G^{(i)}(X,Z)(k_G^{(i)}(Z,Z))^{-1}k_G^{(i)}(Z,X)\Big), \tag{61}$$

where $Z \in \mathbb{R}^{M \times d}$ denotes the inducing points locations, and $X \in \mathbb{R}^{N \times d}$ are data inputs.

In conclusion, the resulting ELBO of the log marginal in Eq.(40) for output $i$ is given by:

$$\text{ELBO}^{(i)} = \text{ELL}^{(i)} - \text{KL}^{(i)} \tag{62}$$

$$= \mathbb{E}_{q\big(f_{0_X}^{(i)}\big)}\left[\log p\big(\mathbf{y}^{(i)}|\mathbb{G}_{\boldsymbol{\Phi}_i}(f_{0_X}^{(i)})\big)\right] + \mathbb{E}_{q\big(\boldsymbol{u}_0^{(i)}\big)}\left[\log \frac{p(\boldsymbol{u}_0^{(i)})}{q(\boldsymbol{u}_0^{(i)})}\right] \tag{63}$$

$$= \sum_n^N \mathbb{E}_{q\big(f_{0_{X_n}}^{(i)}\big)}\left[\log p\left(\mathbf{y}_n^{(i)} \mid \mathbb{G}_{\boldsymbol{\Phi}_i}(f_{0_{X_n}}^{(i)})\right)\right] + \mathbb{E}_{q\big(\boldsymbol{u}_0^{(i)}\big)}\left[\log \frac{p(\boldsymbol{u}_0^{(i)})}{q(\boldsymbol{u}_0^{(i)})}\right], \tag{64}$$

Note that the observation value $\mathbf{y}_n^{(i)}$ only depends on the function evaluation at position $n$. Therefore, minimizing the objective function in the main paper (Eq.(19)) with zero mean equals to:

$$\mathcal{P}_1: \quad \max_{\substack{\boldsymbol{\gamma}^{(i)};\\i=1,2,\ldots,S}} \quad \sum_{i=1}^S \log p(\mathbf{y}^{(i)}), \tag{65}$$

with $\boldsymbol{\gamma}^{(i)} = \{\Theta, \boldsymbol{\alpha}_i, \sigma_i\}$. According to the above inequality in Eq.(40), every output can derive the corresponding lower bound $\text{ELBO}^{(i)}, i = 1, 2, \ldots, S$. Applying the following logic:

$$a < b, c < d: \ a + c < b + d, \quad (a,b,c,d \in \mathbb{R}), \tag{66}$$

we can formulate the variational lower bound of the transformed GMOGP as follows:

$$\sum_{i=1}^S \log p(\mathbf{y}^{(i)}) \geq \sum_{i=1}^S \underbrace{\mathbb{E}_{q\big(f_{0_X}^{(i)}\big)}\left[\log p\big(\mathbf{y}^{(i)}|\mathbb{G}_{\boldsymbol{\Phi}_i}(f_{0_X}^{(i)})\big)\right] + \mathbb{E}_{q\big(\boldsymbol{u}_0^{(i)}\big)}\left[\log \frac{p(\boldsymbol{u}_0^{(i)})}{q(\boldsymbol{u}_0^{(i)})}\right]}_{\text{ELBO}^{(i)}}, \tag{67}$$

Then, instead to solve the optimization $\mathcal{P}_1$ in Eq.(65), we can maximizing its lower bound, namely:

$$\mathcal{P}_2: \quad \max_{\substack{\{\boldsymbol{\gamma}^{(i)},\boldsymbol{\Phi}_i,\boldsymbol{u}_0^{(i)},\boldsymbol{m}_u^{(i)},K_u^{(i)}\};\\i=1,2,\ldots,S}} \quad \sum_{i=1}^S \mathbb{E}_{q\big(f_{0_X}^{(i)}\big)}\left[\log p\big(\mathbf{y}^{(i)}|\mathbb{G}_{\boldsymbol{\Phi}_i}(f_{0_X}^{(i)})\big)\right] + \mathbb{E}_{q\big(\boldsymbol{u}_0^{(i)}\big)}\left[\log \frac{p(\boldsymbol{u}_0^{(i)})}{q(\boldsymbol{u}_0^{(i)})}\right] \tag{68}$$

or $\quad \min_{\substack{\{\boldsymbol{\gamma}^{(i)},\boldsymbol{\Phi}_i,\boldsymbol{u}_0^{(i)},\boldsymbol{m}_u^{(i)},K_u^{(i)}\};\\i=1,2,\ldots,S}} -\left(\sum_{i=1}^S \mathbb{E}_{q\big(f_{0_X}^{(i)}\big)}\left[\log p\big(\mathbf{y}^{(i)}|\mathbb{G}_{\boldsymbol{\Phi}_i}(f_{0_X}^{(i)})\big)\right] + \mathbb{E}_{q\big(\boldsymbol{u}_0^{(i)}\big)}\left[\log \frac{p(\boldsymbol{u}_0^{(i)})}{q(\boldsymbol{u}_0^{(i)})}\right]\right),$

$$\tag{69}$$

which recovers the negative ELBO (NELBO) in the main paper, and the objective function can be further decomposed with locations $n$ as below:

$$\sum_{i=1}^S \sum_n^N \mathbb{E}_{q\big(f_{0_{X_n}}^{(i)}\big)}\left[\log p\left(\mathbf{y}_n^{(i)} \mid \mathbb{G}_{\boldsymbol{\Phi}_i}(f_{0_{X_n}}^{(i)})\right)\right] + \mathbb{E}_{q\big(\boldsymbol{u}_0^{(i)}\big)}\left[\log \frac{p(\boldsymbol{u}_0^{(i)})}{q(\boldsymbol{u}_0^{(i)})}\right] \tag{70}$$

### A.2.3 VARIATIONAL GAP

The gap between the evidence and the ELBO for each output can be measured as follows:

$$\log p(\mathbf{y}^{(i)}) - \text{ELBO}^{(i)}$$

$$= \mathbb{E}_{q\left(f_{K_X}^{(i)}, \boldsymbol{u}_K^{(i)}\right)} \left[\log p(\mathbf{y}^{(i)})\right] - \mathbb{E}_{q\left(f_{K_X}^{(i)}, \boldsymbol{u}_K^{(i)}\right)} \left[\log \frac{p\left(\mathbf{y}^{(i)} \mid f_{K_X}^{(i)}\right) p\left(f_{K_X}^{(i)}, \boldsymbol{u}_K^{(i)}\right)}{q\left(f_{K_X}^{(i)}, \boldsymbol{u}_K^{(i)}\right)}\right] \tag{71}$$

$$= \mathbb{E}_{q\left(f_{K_X}^{(i)}, \boldsymbol{u}_K^{(i)}\right)} \left[\log q\left(f_{K_X}^{(i)}, \boldsymbol{u}_K^{(i)}\right)\right] - \mathbb{E}_{q\left(f_{K_X}^{(i)}, \boldsymbol{u}_K^{(i)}\right)} \left[\log \frac{p\left(\mathbf{y}^{(i)} \mid f_{K_X}^{(i)}\right) p\left(f_{K_X}^{(i)}, \boldsymbol{u}_K^{(i)}\right)}{p(\mathbf{y}^{(i)})}\right] \tag{72}$$

$$= \mathbb{E}_{q\left(f_{K_X}^{(i)}, \boldsymbol{u}_K^{(i)}\right)} \left[\log \frac{q\left(f_{K_X}^{(i)}, \boldsymbol{u}_K^{(i)}\right)}{p\left(f_{K_X}^{(i)}, \boldsymbol{u}_K^{(i)} \mid \mathbf{y}^{(i)}\right)}\right] \tag{73}$$

$$= \text{KL}\left(q\left(f_{K_X}^{(i)}, \boldsymbol{u}_K^{(i)}\right) \| p\left(f_{K_X}^{(i)}, \boldsymbol{u}_K^{(i)} \mid \mathbf{y}^{(i)}\right)\right). \tag{74}$$

The gap is yielded as the Kullback-Leibler divergence between $q(f_{K_X}^{(i)}, \boldsymbol{u}_K^{(i)})$ and $p(f_{K_X}^{(i)}, \boldsymbol{u}_K^{(i)} \mid \mathbf{y}^{(i)})$. This fact forms the basis of the variational inference algorithm for approximate Bayesian inference.

### A.3 GMOGP PREDICTION COMPARISON

For the GMOGP model proposed in Section 3, we can derive the conditional distribution from the joint Gaussian:

$$p(\mathbf{y}^{(i)}, f_X^{(i)}; \boldsymbol{\gamma}^{(i)}) = p(f_X^{(i)}; \Theta, \boldsymbol{\alpha}_i) p(\mathbf{y}^{(i)} | f_X^{(i)}). \tag{75}$$

Correspondingly, the estimates of the output $i$ at a test data $\mathbf{x}_*$ can be calculated from:

$$\text{GMOGP:} \qquad \tilde{f}^{(i)}(\mathbf{x}_*) = k_G^{(i)}(\mathbf{x}_*, X) \underbrace{\left(k_G^{(i)}(X, X) + \sigma_i^2 I_N\right)^{-1} \mathbf{y}^{(i)}}_{\tilde{\boldsymbol{\beta}}}. \tag{76}$$

The counterpart in the classic LMC model admits:

$$\text{LMC:} \qquad \bar{\mathbf{f}}_*[i] = \sum_{j=1}^{S} \sum_{q=1}^{Q} a_{iq} a_{jq} k_q(\mathbf{x}_*, X) \underbrace{\bar{\boldsymbol{\beta}}[(j-1)N + 1 : Nj]}_{\text{N-dimensional vector}}, \tag{77}$$

where $\bar{\mathbf{f}}_*[i]$ denotes the $i^{th}$ element of the LMC prediction vector, and $\bar{\boldsymbol{\beta}} = (K_M(X, X) + \boldsymbol{\Sigma})^{-1} Y$ represents the $NS$-dimensional vector calculated via the high-dimensional gram matrix. Regarding to the output $i$, the prediction only counts on the elements from index $(j-1)N + 1$ to $Nj$.

In contrast, it is noticeable that the first part on the right-hand side of Equations (76) and (77) are both derived composite kernel functions of the test point $\mathbf{x}_*$ and support training points. As shown in the experiments on synthetic data, we find out that little predictive performance gain can be brought by increasing the number of latent independent processes (see Figure 5(b)). Moreover, the LMC with constant coefficients $a_{iq} a_{jq}$ may lack of flexibility and representation ability.

In the context of kernel methods, the predictions are determined by coefficients and a non-linear mapping function in a kernel reproducing Hilbert space. For the above two models, the coefficient $\tilde{\boldsymbol{\beta}}$ and $\bar{\boldsymbol{\beta}}$ both convey the correlated information from other observed output values. Distinct from the LMC, the GMOGP admits less computational complexity and graph learning.

Another alternative view of GMOGP is that it can remedy the cancellation of inter-output transfer in the FICM proposed in Bonilla et al. (2007) with noiseless observations. Since in the noiseless case with a block design, the predictions for output $i$ depend only on the observations $\mathbf{y}^{(i)}$. In other

words, there is a cancellation of information transfer among other outputs. Specifically, given the kernels, the prediction at a test data $\mathbf{x}_*$ for output $i$ can be concluded as:

$$[\overline{\mathbf{f}}(\mathbf{x}_*)][i] = \left[ (K_M(\mathbf{x}, \mathbf{x}) \otimes k(\mathbf{x}_*, X)) (K_M(\mathbf{x}, \mathbf{x}) \otimes k(X, X))^{-1} Y \right][i] \tag{78}$$

$$= \left[ \left\{ \left( K_M(\mathbf{x}, \mathbf{x}) (K_M(\mathbf{x}, \mathbf{x}))^{-1} \right) \otimes \left( k(\mathbf{x}_*, X) (k(X, X))^{-1} \right) \right\} Y \right][i] \tag{79}$$

$$= k(\mathbf{x}_*, X) (k(X, X))^{-1} \mathbf{y}^{(i)}. \tag{80}$$

Here, the $[\overline{\mathbf{f}}(\mathbf{x}_*)][i]$ represents the $i^{th}$ element of the vector-valued prediction. Compared with the GMOGP in Eq.(76) in the noiseless version, the information in other outputs can be implied from the inner product based attention coefficients enrolled in the aggregated kernel function.

## A.4 JOINT DISTRIBUTION OF MULTIPLE OUTPUTS

From the main paper, the joint distribution of all $S$ outputs defined by the specific graph structure can be formulated as:

$$p(f_X^{(1)}, f_X^{(2)}, \ldots, f_X^{(S)}) = p(f_X^{(i)}|\mathrm{pa}_i)p(\mathrm{pa}_i) \tag{81}$$

$$= p(f_X^{(i)}|\mathrm{pa}_i) \prod_{j \in \mathrm{pa}_i} p(f_X^{(j)}) \tag{82}$$

In the GMOGP model, we model the target output with the following conditional distribution:

$$p(f_X^{(i)}|\mathrm{pa}_i) = \mathcal{N}\Big( f_X^{(i)} \Big| \sum_{j \in \mathrm{pa}_i} \alpha_{i,j} f_X^{(j)} + \boldsymbol{m}_i, k_{\boldsymbol{\theta}_i}(X, X) \Big), \ \ i \in \mathcal{I}, \tag{83}$$

and $p(f_X^{(j)}) = \mathcal{N}(\boldsymbol{m}_j, k_{\boldsymbol{\theta}_j}(X, X)), j \in \mathrm{pa}_i$. After the model training, given the learned parameters, we can obtain the corresponding joint distribution from Eq.(82). While in the TGMOGP, since we transform the GMOGP prior with the marginal flow, the TGMOGP prior becomes:

$$p_{\boldsymbol{\gamma}^{(i)}, \boldsymbol{\Phi}_i}\big( f_{K_X}^{(i)} | \mathbb{G}, X \big) = p_{\boldsymbol{\gamma}^{(i)}}\big( f_X^{(i)} \big) \prod_{k=0}^{K-1} \left| \det \frac{\partial \mathbb{G}_{\phi_k^{(i)}}\big( f_{k_X}^{(i)} \big)}{\partial f_{k_X}^{(i)}} \right|^{-1}. \tag{84}$$

Under the equality derived in Eq.(27), the invertible transformations on the target node $i$ can be represented as:

$$\mathbb{G}_{\boldsymbol{\Phi}_i}(f_X^{(i)}) = \mathbb{G}_{\boldsymbol{\Phi}_i}\left( \sum_{j \in \mathrm{pa}_i} \alpha_{i,j} f_X^{(j)} + \boldsymbol{m}_i + \boldsymbol{\psi}_i \right). \tag{85}$$

Given the parent nodes $\mathrm{pa}_i$, we have

$$p(\mathbb{G}_{\boldsymbol{\Phi}_i}(f_X^{(i)})|\mathrm{pa}_i) = p(f_X^{(i)}|\mathrm{pa}_i) \prod_{k=0}^{K-1} \left| \det \frac{\partial \mathbb{G}_{\phi_k^{(i)}}\big( f_{k_X}^{(i)} \big)}{\partial f_{k_X}^{(i)}} \right|^{-1}. \tag{86}$$

Therefore, the resulting joint distribution can also be deduced with learned coefficients and hyper-parameters.

## A.5 PARETO OPTIMALITY

Pareto optimality is a foundational concept in the optimization community. In single objective problems, the Pareto optimal solution is unique. In the context of the multi-objective optimization, the multiple objectives need to be optimized simultaneously, looking for a set of Pareto optimal solutions. This process is called multi-objective optimization (Censor, 1977). In short, the Pareto optimal solution is a set of 'non-inferior' solutions in the objective space, and defines a boundary beyond that none of the objectives can be improved without sacrificing at least one of the other objectives.

The benefits of solving the multi-output regression in a multi-objective problem (MOO) instead of the classic high-dimensional optimization considered in the MOGP methods are discussed here. Since the kernel hyperparameters $\Theta$ are shared among all different outputs to achieve information exchange, we can build an MOO problem,

$$\boldsymbol{F}(\Theta) = [\mathcal{L}^{(1)}(\Theta), \mathcal{L}^{(2)}(\Theta), \ldots, \mathcal{L}^{(S)}(\Theta)]^T, \tag{87}$$

with S objectives in the following form:

$$\mathcal{L}_{\boldsymbol{\gamma}^{(i)}}^{(i)} \propto \left\{ (\tilde{\mathbf{y}}^{(i)})^T \left( k_G^{(i)}(X, X) + \sigma_i^2 I_N \right)^{-1} \tilde{\mathbf{y}}^{(i)} + \log \left| k_G^{(i)}(X, X) + \sigma_i^2 I_N \right| \right\}. \tag{88}$$

There exists competition among the multiple objectives, as the outputs correspond to multiple data sources and distinct hyperparameters. The illustration of the competition between two objectives is given in Figure 6, where the Pareto optimal solutions represent the points at the Pareto front.

Without loss of generality, the Pareto optimal solutions of the kernel hyperparameters are better than other feasible solutions, and can obtain more stable performance compared to those learned by a high-dimensional optimization problem built in state-of-the-art (SOTA) MOGP methods. Furthermore, the objective functions for target outputs are parameterized by unique attention coefficients, all of which are simultaneously minimized to zero.

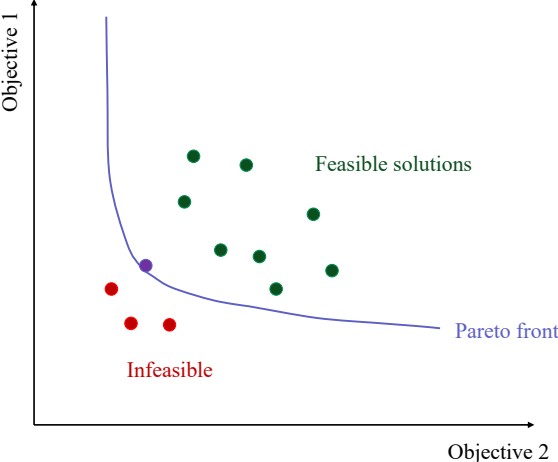

Figure 6: The description of the Pareto optimal solutions with respect to the multi-objective problem.

## B  ADDITIONAL DETAILS ON EXPERIMENTS

In this section, we explicit the detailed experiment settings, learning parameters for all competitive models, and extra experiment results of various real-world datasets.

### B.1  LEARNING PARAMETERS

In the competing methods, the primitive kernel function is selected to be the Squared Exponential (SE). The hyperparameters contain a length-scale $l$, an output-scale/signal variance $c$, namely

$$k_{\text{SE}}(\mathbf{x}, \mathbf{x}') = c \cdot \exp \left( - \|\mathbf{x} - \mathbf{x}'\|^2 / l^2 \right). \tag{89}$$

The detailed learning parameters are specified in Table 4, where $l_{NF}$ denotes the number of stacked transformations in the flow (specific parameters of the flow are listed in section B.2), and the $V_m$ represents the quantity of variational parameters (including the inducing points, the mean and covariance in the variational distribution). Also, the $Q$ denotes the number of latent independent Gaussian processes used in the LMC variants, and the $S$ represents the number of outputs.

Table 4: The detailed learning parameters for competing models.

| Acronym | Learning parameters (style and size) |
|---------|---------------------------------------|
| [1] SOGP | noise: $S$, length-scale: $S$, output-scale: $S$, mean: $S$ |
| [2] LMC | noise: $S+1$, mean: $S$, coefficients: $SQ$ , length-scale: $Q$, output-scale: $Q$ |
| [3] FICM | noise: $S+1$, mean: $S$, raw-variance: $S(S+1)/2$ , length-scale: 1 |
| [4] GPRN | noise: $S+1$, mean: $S$ , length-scale: $Q+1$, output-scale: $Q+1$ |
| [5] GMOGP | noise: $S+1$, mean: $S$, attention: $S^2+S$, length-scale: $S$, output-scale: $S$ |
| [6] TGMOGP | noise: $S+1$, mean: $S$, attention: $S^2+S$, length-scale: $S$, output-scale: $S$, $4l_{NF}$, $V_m$ |

## B.2 MARGINAL FLOWS

In the experiments, we stack $K = 3$ or 4 layers of a composite Sinh-Archsinh flow with Affine flow (SAL), which can be formulated as below:

$$f_{K_X}^{(i)} = c^{(i)} \left( \sinh \left( b^{(i)} \cdot \operatorname{arcsinh} \left( f_{K-1_X}^{(i)} \right) - a^{(i)} \right) \right) + d^{(i)}. \tag{90}$$

For the output $i, i \in \mathcal{I}$, each layer in the flow has four free parameters, i.e., $a^{(i)}, b^{(i)}, c^{(i)}, d^{(i)} \in \mathbb{R}$. Some commonly used and available formulas of the flow are listed in the following Table 5. More discussions and validity analysis can be found in (Rios & Tobar, 2019).

Table 5: The list of common marginal flow types with input $f_X^{(i)}$. The parameters are constrained so that each individual flow is strictly increasing/decreasing functions.

| Flow | Forward | Inverse | Parameters |
|------|---------|---------|------------|
| Log | $\log \left( f_{0_X}^{(i)} \right)$ | $\exp(f_{K_X}^{(i)})$ | $-$ |
| Softplus | $\log \left( \exp \left( f_{0_X}^{(i)} \right) + 1 \right)$ | $\log \left( \exp \left( f_{K_X}^{(i)} \right) - 1 \right)$ | $-$ |
| Affine | $a + b \cdot f_{0_X}^{(i)}$ | $(f_{K_X}^{(i)} - a)/b$ | $a, b \in \mathbb{R}$ |
| Sinh-Archsinh | $\sinh \left( b \cdot \operatorname{arcsinh} \left( f_{0_X}^{(i)} \right) - a \right)$ | $\sinh \left( \frac{1}{b} \left( \operatorname{arcsinh} \left( f_{K_X}^{(i)} \right) + a \right) \right)$ | $a, b \in \mathbb{R}$ |
| Boxcox | $\frac{1}{\lambda} \left( \operatorname{sgn} \left( f_{0_X}^{(i)} \right) \left| f_{0_X}^{(i)} \right|^{\lambda} - 1 \right)$ | $\operatorname{sgn} \left( \lambda \cdot f_{K_X}^{(i)} + 1 \right) \left| \lambda \cdot f_{K_X}^{(i)} + 1 \right|^{\frac{1}{\lambda}}$ | $\lambda > 0$ |
| TanH | $a \tanh \left( b \left( f_{0_X}^{(i)} + c \right) \right) + d$ | $-$ | $a, b, c, d \in \mathbb{R}$ |

## B.3 REAL WORLD EXPERIMENTS

This section shows more detailed experiment results of the multi-output regression tasks across various real datasets. The data descriptions and specified tasks are demonstrated in Table 6.

Table 6: Description of the applied real datasets and multi-output regression tasks.

| Dataset | $N_{\text{train}}$ | $N_{\text{test}}$ | $D_X$ | $S$ | Tasks |
|---------|--------|--------|-------|-----|-------|
| JURA | 249 | 100 | 2 | 3 | Cadmium, nickel, and zinc concentration prediction in Jura |
| EEG | 256 | 100 | 1 | 7 | Extrapolation of sampled signals from electrodes on scalps |
| TRAFFIC | 500 | 172 | 1 | 6 | Traffic time-series prediction for three adjacent cells |
| ECG | 6000 | 2000 | 1 | 5 | Extrapolation of signals from electrodes on navel and head |
| KUKA | 12235 | 5325 | 21 | 7 | Torques prediction of seven joints for an lightweight arm |
| SARCOS$_1$ | 2000 | 4449 | 21 | 4 | Torques prediction of the 2,3,4,7 joints for an anthropomorphic arm |
| SARCOS$_2$ | 20000 | 4449 | 21 | 4 | |

### B.3.1 EEG DATA FITTING FOR THE FIRST OUTPUT

The main results of the proposed GMOGP models on the EEG dataset are shown in the main paper. Here, we want to draw the predictions of the output regarding to the electron $F_0$ and show more data

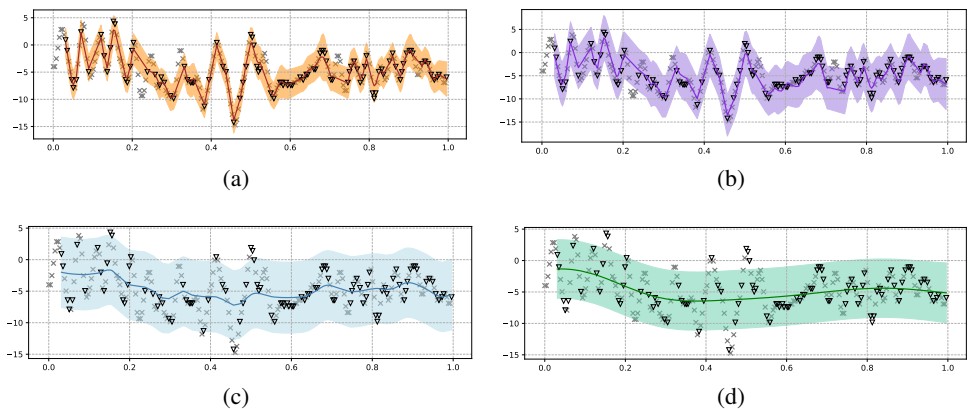

Figure 7: The predictive performance of the EEG data with respect to the signal of electron $F_0$. Sub-figures show the fitting results from different methods: (a) GMOGP, (b) FICM, (c) LMC, (d) SOGP, where the cross items represent the training data points, and the inverted triangles are the locations of the 100 test points.

fitting details. In Figure 7, we can see the GMOGP outperforms the other competing methods with lower predictive uncertainty.

## B.4 KUKA Predictive Performance for the Multiple Outputs

In Figure 8, we show the test RMSE of each output, and compare the average RMSE on the seven outputs. The standardized KUKA data has close mean, median, and mode, which may be better described by the Gaussian predictive distributions.

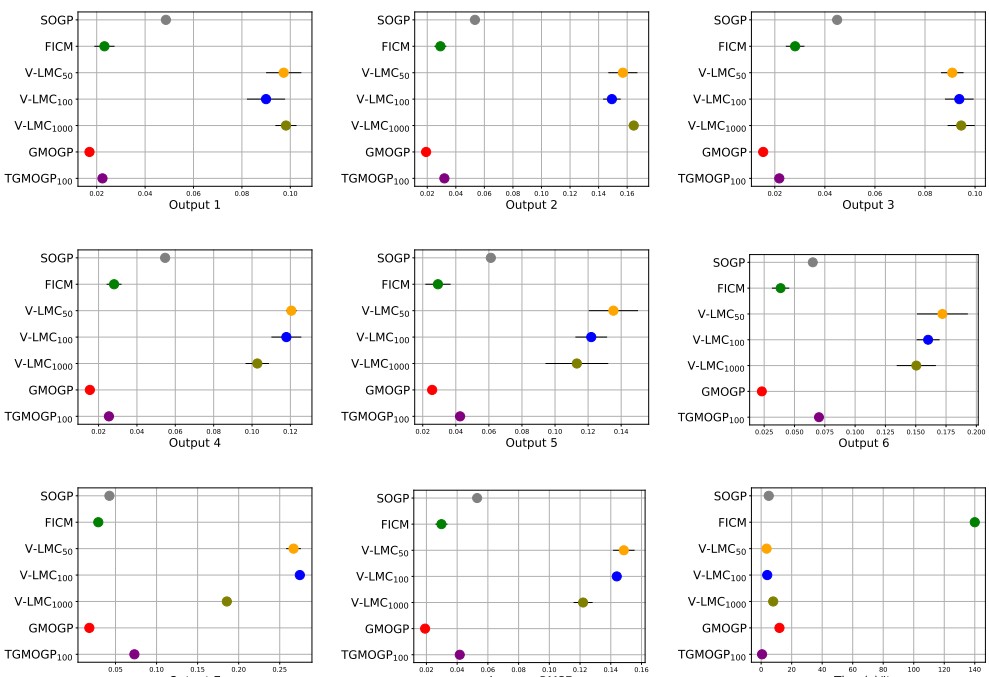

Figure 8: The sub-figures show the test RMSE for output 1 to output 7. The last two show the average RMSE and time cost per iteration.

### B.4.1 TRAFFIC DATA EXTRAPOLATION

In this section, we test our proposed GMOGP models on the real time-series dataset:

**Traffic data:** The data containing 28 days of traffic measurements of 3 cells are collected by communication base stations in China. For each cell, we gathered two traffic time-series data (the amount of traffic that has been accumulated per hour in 28 days). Each traffic data contains 672 samples. We use the first 500 samples for training and the last 172 for testing. This multi-output regression task can be modeled as a GMOGP with six outputs, and the data are drawn in Figure 9.

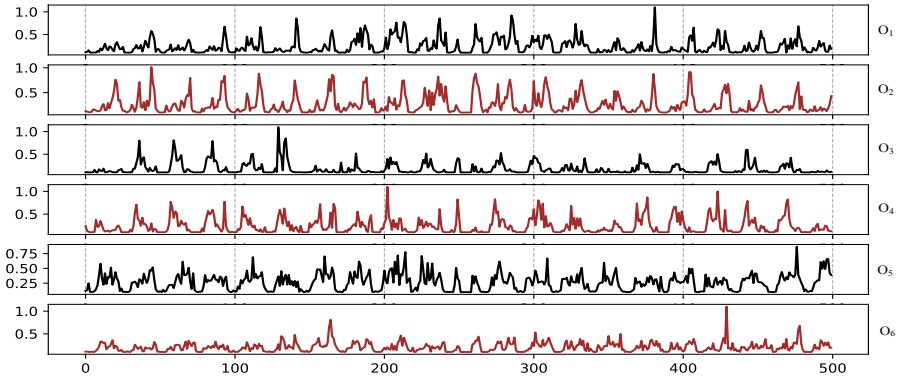

Figure 9: The description of the traffic data from 3 neighbored cells in 28 days. The lines paired with black and red from top to bottom represent the traffic data in the same cell of different orientations.

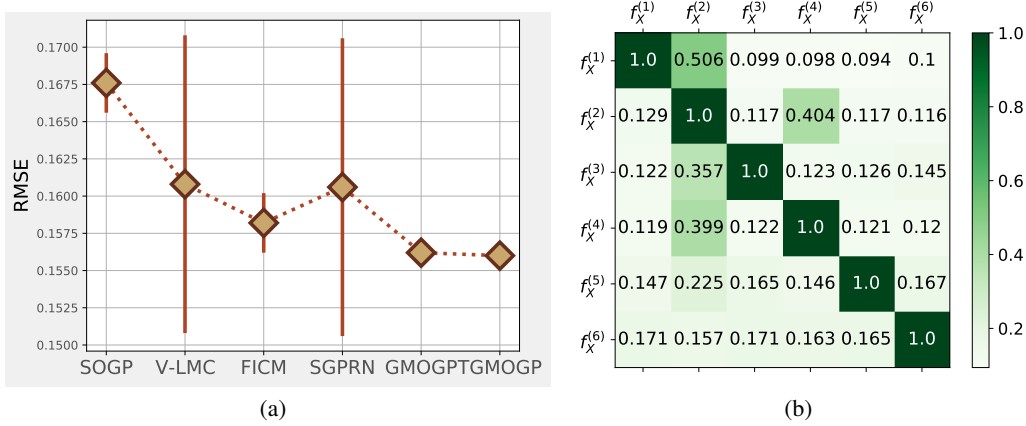

(a)                                                                      (b)

Figure 10: Traffic data predictive results, namely, (a) average RMSE of different methods, and (b) learned attention coefficient values $\alpha$.

Figure 10(a) shows the average test RMSE on the traffic dataset over five runs. The results demonstrate that the GMOGP-based models outperform other MOGP methods and the SOGP model. Since the real data is non-smooth and contains zero values, the isolated SOGP model can hardly extrapolate well with insufficient model flexibility brought by a primitive kernel. According to the attention coefficients learned by the GMOGP model in Figure 10(b), we can interpret graphical representations and construct the joint distribution of all nodes. Intuitively, we can find out the output 1 has strong conditional dependence with the output 2, and it also can be noticed by the coefficient $\alpha_{1,2}$. In fact, the $1^{st}$ and the $2^{rd}$ outputs are collected in the same cell with similar trends in practice. However, the other output with different envelope and regularity can provide little useful information when predicting the first output. Comparing to the classic MOGP models that correlate all outputs without selection, our GMOGP can provide an efficient alternative for multi-output regression.

## C    LIMITATIONS, EXTENSIONS, AND FUTURE WORK

The main limitations of the GMOGP can be mitigated through various avenues:

1. Interpretation of the attention coefficients: The attention coefficients signify the dependence among nodes and graph structures in this paper. However, interpretations of the coefficients learned by the TGMOGP and SOTA graph attention networks are vagued and anticipated in Explainable Artificial Intelligence (XAI) field.

2. Dynamic dependence measure: The interaction or dependence dynamically changes with inputs.

3. Extensions of online learning frameworks and graph construction schemes: Opportunities exist for extending the GMOGP to online learning frameworks and enhancing graph construction schemes in an online setting.

4. Adaptation of attention mechanism/scoring function/dependence measure for different tasks: Consideration of diverse attention mechanisms, scoring functions, and dependence measures tailored to specific tasks in practice.

5. Exploration of different aggregation models (Eq. (16)).

6. Exploration of different objective functions and weighting schemes tailored for varied applications.

In the future, exploration on the use of GMOGP with various types of adaptive/input-dependent covariance structures and flows would be instructive. Additionally, extending the GMOGP to equip data-dependent mean function, model multiple fidelity outputs, and facilitate sequential forecasting would be of great interest. We hope the GMOGP will inspire further research into explainable networks, bridging benefits from learning models and statistics.

