# OpenReview forum: "Graphical Multioutput Gaussian Process with Attention"
_ICLR.cc/2024/Conference — ICLR 2024 spotlight_

### Official Review · Reviewer_X7C1 · 2023-10-29

**Soundness:** 2 fair
**Presentation:** 2 fair
**Contribution:** 2 fair
**Rating:** 6
**Confidence:** 4

**Summary:**

To alleviate the computational complexity and expressiveness of MOGP models, this article proposes a multioutput regression model where each output is modelled as a (marginal) single-output GP, then, these variables are "tied" together as nodes of a graph (eq 11). The adjacency of the nodes is then learnt via an attention mechanism (Sec 3.2). The authors also proposed transformed (non-linear) and sparse variants of their method. The article features synthetic and real-world experiments against some MOGP models.

**Strengths:**

The idea of connecting GPs with graphs and attention networks is of interest for the community, a number of researchers will be interested in this interface.

**Weaknesses:**

Although the general idea of building a graph where nodes are GPs is attractive and to some extent promising, the concept is not properly exploited in the article. In this regard, the most relevant weaknesses of the paper are:

- The proposed dependence structure between an output node and its parents is Gaussian (eqs 12 and 13), which in turn results in an overall (MO)GP. Therefore, the proposed model could be understood as a (standard) MOGP with a specific kernel. In addition to the conceptual impact of this fact, there is also a practical consequence: It is unclear if the exhibited experimental superiority of the proposal comes from the introduced graph idea or the fact the there is an implicit, more expressive, kernel being used. From the paper is not possible to clarify this because the proposal is only compared empirically to classic kernels for MOGP (e.g., LMC), leaving behind the MO spectral mixture, deep kernels, cross spectral mixture, etc.

- The proposed model introduced the (hyper) parameters \alpha as weights learned via attention. These hypers have a direct effect on the linear combination of the mean of the marginal GP which models each node. Though the intuition of attention is that it represents cross relationships among variable of interest, there is no reason to assume that they _linearly_ affect the output. In general, attention networks have an MLP stage after the attention stage, where the MLP "discovers" the role of the attention weights.

- The authors claim that standard MOGP have _enlarged searching space with countless local optima_. This is general not true, in GPs (and MOGPs), hyperparameters are usually of very low dimensionality (in particular in the LMC considered in this work as benchmark).

- Diagrams in Figures 3 and 4 are not explained, neither in the body of the paper nor in the Figures' captions. Therefore, they do not contribute to a better understanding of the paper.

- In one of the experiment, MOGP performed worse than single-output GP. This is clearly a training artifact: If SOGP are a restriction of MOGP (same kernel), the latter has to perform at least equal or better than the former, unless MOGP is incorrectly trained.

- The purpose of Experiment 5.1 is not clear: synthetic data are not Gaussian, meaning that the measured performance of the methods considered is not indicative of their applicability for the general case or even for the Gaussian case (a hypothesis shared by all models).

- Given the proposed model builds upon a direct effect among  nodes, how can one deal with the missing data case?

- There are relevant works in the literature that are not mentioned in this paper. A model that is particularly similar is the Semiparametric latent factor model [SLFM](https://proceedings.mlr.press/r5/teh05a.html), which builds an MOGP by linearly combining a set of single-output GPs (rather than relating them through a graph as done in this paper). Also, there have been a number of expressive kernels proposed in the last decade, but this paper only considers LMC (one of the first approaches) and GPRN.

**Questions:**

Please refer to the comments in the previous section

---

> ### Author Response · Authors · 2023-11-19
> **Thank you for your feedback! please consider our response to your questions - Part 1.**
>
> Thank you for your review! We would like to underscore that our work introduces a novel MOGP framework, offering extra flexibility and graphical representations. We address your key concerns below.
>
> ***
> >Q1. "The proposed dependence structure between an output node and its parents is Gaussian (eqs 12 and 13), which in turn results in an oveall (MO)GP. Therefore, **the proposed model could be understood as a (standard) MOGP with a specific kernel**. In addition to the conceptual impact of this fact, there is also a practical consequence: **It is unclear if the exhibited experimental superiority of the proposal comes from the introduced graph idea or the fact the there is an implicit, more expressive, kernel being used**. From the paper is not possible to clarify this because the proposal is only compared empirically to classic kernels for MOGP (e.g., LMC), leaving behind the MO spectral mixture, deep kernels, cross spectral mixture, etc."
>
> The proposed model cannot be understood as a standard MOGP with a specific kernel due to a crucial reason:
> * We interpret the correlation with **asymmetric measure**. Any specific MOGP kernel must be symmetric. In our approach, the interaction (correlation) between different outputs is quantified by the attention mechanism (denoted as $\alpha_{ij}$). Consequently, our GMOGP introduces an asymmetric interaction measure ($\alpha_{ij}\neq \alpha_{ji}$), offering enhanced flexibility.
>
> As a result, there is **no necessity to construct the large kernel matrix (Eq. (5))**, thereby alleviating computational and storage burdens. In this paper, we explored an alternative graphical MOGP model structure and learning framework.
>
> By introducing the graphical model, we can derive a more flexible GP prior (even non-Gaussian), flexible interaction measurements, an efficient learning framework, and graphical representations. This heightened model flexibility and improved representation enable us to achieve experimental superiority.
>
> In our experiments, we aim to discern the prediction performance between the proposed graphical MOGP and the classic MOGP **framework** using the same base kernels. Additionally, our GMOGP accommodates other expressive kernels (spectral mixture, deep kernel) for further exploration.
>
> >Q2. The proposed model introduced the (hyper) parameters \alpha as weights learned via attention. These hypers have a direct effect on the **linear combination** of the mean of the marginal GP which models each node. Though the intuition of attention is that it represents cross relationships among variable of interest, there is no reason to assume that they **linearly affect the output**. In general, **attention networks have an MLP stage after the attention stage**, where the MLP "discovers" the role of the attention weights.
>
> Starting from simpler concepts and progressing to more complex ones, we initially demonstrate how a multivariate Gaussian can be represented as a directed graph, corresponding to a linear Gaussian model (linear combination). Subsequently, **in Section 3.4, we address "linearly affect" by introducing the marginal flow**. Specifically, the target node in the TGMOGP undergoes transformation via the non-linear Sinh-Archsinh (SAL) function (denoted by ${\mathbb{G}}_{\phi_k}(\cdot)$ ) in our experiments. See Eq.(52) in Appendix A.
>
> In the realm of graph attention networks, the MLP stage is indispensable for achieving the final prediction. In contrast, the prediction in the GMOGP is derived through GP inference, not following an 'end-to-end' approach. Furthermore, we can apply **K layers of non-linear transformations ('deeper MLP stage'), leading to a more adaptable GP prior/posterior**. In our experiments, we implement 3-layer SAL transformations. Detailed information about the TGMOGP model and other non-linear transformation functions is provided in Section 3.4 and Appendix B. We emphasized this insight in our revised paper.

---

> > ### Author Response · Authors · 2023-11-20
> > **Thank you for your feedback! please consider our response to your questions - Part 2.**
> >
> > >Q3. The authors claim that standard MOGP have **enlarged searching space** with countless local optima. This is general not true, in GPs (and MOGPs), hyperparameters are usually of very low dimensionality (in particular in the LMC considered in this work as benchmark).
> >
> > The enlarged searching space, as compared to the SOGP, is a consequence of employing a matrix-valued kernel function and a vector-valued reproducing kernel Hilbert space (RKHS). In both SOGP and the proposed GMOGP, we search for scalar-valued kernel functions in scalar-valued RKHSs. Considering the definition of a kernel function in the inner product space, where the kernel $k(x,x’)=<\phi(x),\phi(x’)>$, the dimension of random feature vectors $\phi(x)$ with respect to the MOGP is higher than the scaler-output GP.
> >
> > For an in-depth explanation of vector-valued RKHS and kernel methods, please refer to [1]. Apart from the high-dimensional searching space, achieving comparable performance to isolated GPs in the LMC variants requires hundreds of latent independent GPs. As demonstrated in [2], the MOGP can outperform isolated SOGPs when the number of latent GPs exceeds 200. Consequently, **classic LMC involves hundreds of vector-valued RKHSs**. We have clarified this point in our updated version.
> >
> > >Q4. Diagrams in Figures 3 and 4 are not explained, neither in the body of the paper nor in the Figures' captions. Therefore, they do not contribute to a better understanding of the paper.
> >
> > Figure 3 is pivotal, illustrating the **fundamental framework distinctions between the GMOGP and standard MOGPs**. In this representation, each output follows its unique workflow, contributing specific information and converging with other dependent information through shared kernel hyperparameters. This adaptable framework **seamlessly extends to distributed learning schemes without necessitating additional approximations or assumptions**. Figure 4 encapsulates all variable dependencies within the GMOGP. We added more descriptions in the updated version.
> >
> > >Q5. In one of the experiment, MOGP performed worse than single-output GP. This is clearly a training artifact: **If SOGP are a restriction of MOGP (same kernel), the latter has to perform at least equal or better than the former**, unless MOGP is incorrectly trained.
> >
> > With limited data samples, the MOGP often leverages additional information to yield more accurate predictions. However, in scenarios with an ample supply of training samples, the MOGP's performance surpassing the SOGP is contingent upon meeting specific prerequisites:
> > 1. **A proper interaction measure** is crucial for selecting correlative information from other outputs.
> > 2. **Adequate MOGP model flexibility** and representation ability are necessary, enabling the model to learn the unique information of each output and effectively aggregate useful information from dependent outputs.
> > 3. The selection of **appropriate training objective functions and learning schemes** is vital.
> >
> > Increasing the number of latent GPs, denoted as Q, enhances the model flexibility of LMC variants. Notably, experiments in Table 1 of the paper [2] highlight that the MOGP outperforms isolated SOGPs when Q exceeds 200. However, determining the optimal Q for diverse tasks is challenging due to the inherent trade-off between model flexibility and computational efficiency. In our experiments, we pragmatically set Q equal to the number of outputs, aligning with common practices.
> >
> > >Q6. **The purpose of Experiment 5.1 is not clear: synthetic data are not Gaussian**, meaning that the measured performance of the methods considered is not indicative of their applicability for the general case or even for the Gaussian case (a hypothesis shared by all models).
> >
> > While GPs prove effective, their efficacy is **limited** by Gaussian assumptions in the prior, likelihood, and posterior. To overcome this constraint, several non-Gaussian GP models have been proposed, maintaining a Gaussian copula and addressing model mismatches when handling real data with unknown distributions.
> >
> > In alignment with this approach, our paper proposes a transformed GMOGP tailored for real non-Gaussian data. Experiment 5.1 was designed to assess the **generalization capability** of standard MOGPs on non-Gaussian data and to evaluate the effectiveness of the TGMOGP.

---

> ### Author Response · Authors · 2023-11-20
> **Thank you for your feedback! please consider our response to your questions - Part 3.**
>
> >Q7. Given the proposed model builds upon a direct effect among nodes, how can one deal with the **missing data** case?
>
> For low missing rates, standard imputation or interpolation methods suffice. In the case of unbalanced data, where some outputs have significant missing entries, we discard the missing data since the **GMOGP models each output separately with shared kernel hyperparameters**.
>
> The vanilla GMOGP, grounded in Equation (13), obtains predictions through inference, diverging from the simple elementwise weighted summation as in attention networks. Proposition 3.1 asserts that the GP prior for **each output depends solely on its own data**, facilitating the effective handling of unbalanced data per output.
>
> Dealing with unbalanced data requires an additional weight matrix for transforming attention inputs to a uniform dimension, a common practice in attention networks.
> The updated version includes more concise descriptions, clarifications, and insights into the model's limitations.
>
> >Q8. There are relevant works in the literature that are not mentioned in this paper. A model that is particularly similar is the Semiparametric latent factor model **SLFM**, which builds an MOGP by linearly combining a set of single-output GPs (rather than relating them through a graph as done in this paper). Also, there have been a number of expressive kernels proposed in the last decade, but this paper only considers LMC (one of the first approaches) and GPRN.
>
> The SLFM is a **instance** of the GPRN and the LMC (see (Bonilla et al., 2007), (Wilson et al., 2011), (Liu et al., 2018), etc.). However, the **GPRN already outperforms SLFM in the Jura dataset** (Wilson et al., 2011). Advanced MOGP models, such as LMC, SLFM, GPRN, convolutional LMC, etc., necessitate the construction of large covariance matrices, each characterized by distinct matrix-valued kernel functions.
>
> In contrast, the proposed GMOGP **departs from the need for matrix-valued kernel design**, presenting a unique paradigm in MOGPs. It provides an efficient framework with enhanced flexibility and representations, **accommodating more expressive kernels**.
>
> In recent experiments, we introduced convolutional MOGP (CMOGP [3]), a non-separable model unlike LMC variants. See the table for predictive results. However, CMOGP encounters memory limitations when applied to the full SARCOS dataset.
>
>
> |  | SOGP | CMOGP|V-LMC |SGPRN |FICM |GMOGP|TGMOGP|
> |------------------|------------------|------------------|------------------|------------------|------------------|------------------|------------------|
> | Synthetic data  | 0.5653$\pm$0.0023 | 0.5539$\pm$0.0089  |0.5917$\pm$0.0096   |0.5819$\pm$0.0207 |0.5544$\pm$0.0046 |0.5541$\pm$0.0054  |0.5343$\pm$0.0023 |
> | Jura   | 0.605$\pm$0.01|0.429$\pm$0.01 |0.443$\pm$0.01 |0.438$\pm$0.02 |0.394$\pm$0.05| 0.376$\pm$0.01 |0.382$\pm$0.01  |
> ***
>
> $ $
> ***
> * [1]  Mauricio A Alvarez, Lorenzo Rosasco, and Neil D Lawrence. "Kernels for vector-valued functions: A review." Foundations and Trends in Machine Learning, 4(3):195–266, 2012.
> * [2]  Bruinsma, Wessel, et al. "Scalable exact inference in multi-output Gaussian processes." International Conference on Machine Learning. PMLR, 2020
> * [3]  M.A. Álvarez and N.D. Lawrence, "Sparse Convolved Multiple Output Gaussian Processes", Advances in Neural Information Processing Systems 21, 2009

---

> > ### Author Response · Authors · 2023-11-23
> >
> > Dear respected reviewer,
> >
> > Thanks again for your valuable review comments that helped improve the quality of our draft significantly.
> >
> > Please let us know if our answers resolved your questions/concerns.
> >
> > Many thanks!

---

> > > ### Comment · Reviewer_X7C1 · 2023-12-03
> > >
> > > Dear Authors,
> > >
> > > Thanks very much for carefully replying to my comments and incorporating my recommendations. After reading your reply and going through the interaction with other reviewers, I find this paper has improved and so has my understanding of the contribution. I believe this paper will be a good fit for the conference, therefore, I have increased my score and recommend acceptance.

---

### Official Review · Reviewer_pYyL · 2023-10-30

**Soundness:** 3 good
**Presentation:** 4 excellent
**Contribution:** 3 good
**Rating:** 8
**Confidence:** 4

**Summary:**

This paper delves into the multiple output regression problem, a significant challenge in both machine learning and statistics. It focuses on predicting multiple target variables concurrently. Multiple output Gaussian process regression (MOGP) is an extension of Gaussian process regression (GP) tailored for handling multiple target variables. While MOGP methods provide excellent solutions with uncertainty quantification, they are hindered by substantial computational complexity and storage requirements. The paper introduces an innovative approach to address these complexities and enhance prediction quality by incorporating graphical models and an attention mechanism.

**Strengths:**

In general, the paper has been well-written, particularly in the first two sections where the authors aptly define the problem, establish motivation, and present existing baselines. The foundational concept of addressing the MOGP problem through the lens of graphical models is intriguing, as it offers a means to elucidate the interplay between various target variables.

The primary contribution of this paper, which involves leveraging distributed learning and attention mechanisms, is commendable given the capabilities of these models. The experiments and numerical results underscore a significant enhancement in prediction quality and execution time when compared to other baseline methods.

**Weaknesses:**

The current manuscript still exhibits several weaknesses. Notably, in certain sections, the paper's coherence is lacking, making it challenging for readers to follow. Mathematical concepts, in particular, suffer from inadequate definitions. For instance, the paper introduces $f^{(i)}_X$ as a latent variable representing the GP functions for a specific target variable i. However, it later employs $f^{(i)}_X$ to denote the distribution of a GP conditioned on its parents (as seen in Eq. 11 and Eq. 13). Given that prior and conditional distributions hold paramount importance in this work, it is imperative that related variables are precisely defined, and vague expressions are avoided.

Additionally, a considerable portion of the explanations provided in Section 2 concerning directed graphs appears to be redundant. The authors delve into the intricacies of directed graphs and the definition of a Bayesian network, yet the second paragraph on page 4 reveals that the graph relating the GPs in this paper is a fully connected undirected graph (as indicated by the definition of $pa_i$ for all output functions). It begs the question of why the authors did not emphasize this type of graph from the outset. There are several undirected graph models that can be pertinent when dealing with Gaussian variables, including Gaussian graphical models, pararnormal/nonparametric graphical models, and functional graphs.

Indeed, the description of the distributed Gaussian process (DGP) in the paper is lacking. The conventional DGP model, as proposed by Deisenroth and Ng in 2015, operates under two fundamental assumptions. Firstly, it assumes that all GP experts share the same hyperparameters during the training phase, which serves as a mechanism to address overfitting. Secondly, it assumes perfect diversity between the target variable GPs. However, this second assumption does not align with the scope of MOGP, as interactions between GPs are a crucial aspect of the model. The paper also falls short of providing a clear explanation of how the distributed framework depicted in Figure 4 functions.

To mitigate computational costs, the model introduces DGP, which can also be applied to conventional MOGP. However, the paper falls short in discussing why this solution is faster than the other baselines. Both the attention mechanism and non-Gaussian transformation inherently raise computational costs, making it essential to have a comprehensive and detailed discussion on this issue within the paper. This is particularly important because it is one of the claims made in the paper. Regrettably, the authors only touch upon this matter in the experiments section, which is not sufficient to provide a thorough understanding of the computational efficiency of their approach.

A similar issue pertains to the prediction quality. The proposed solution integrates a combination of existing methods within a unified framework. However, the question arises: why does this method enhance prediction quality and surpass other baselines? While the experiments demonstrate a significant improvement, the underlying reasons have not been sufficiently elucidated within the paper.

**Questions:**

Please see the discussions in the Weaknesses section.

---

> ### Author Response · Authors · 2023-11-20
> **Thanks for your review! please consider our response to your concerns - Part 1**
>
> Thank you for your feedback. We wish to highlight that our **GMOGP can seamlessly extend to a distributed MOGP framework without compromising interaction information**. We trust that the responses effectively address your concerns.
> ***
> >Q1. The current manuscript still exhibits several weaknesses. Notably, in certain sections, the paper's coherence is lacking, making it challenging for readers to follow. Mathematical concepts, in particular, suffer from inadequate definitions. **For instance, the paper introduces $ f_X^{(i)} $ as a latent variable representing the GP functions for a specific target variable i. However, it later employs $ f_X^{(i)} $
>  to denote the distribution of a GP conditioned on its parents (as seen in Eq. 11 and Eq. 13)**. Given that prior and conditional distributions hold paramount importance in this work, it is imperative that related variables are precisely defined, and vague expressions are avoided.
>
> In Eq. (11), $ f_X^{(i)} $ represents the target node in the specific directed graph.
> Preceding Eq. (11), each node, including the target node, is explicitly defined as a SOGP evaluated at finite inputs X. Hence, $ f_X^{(i)} \in \mathbb{R}^N$ and **follows a multivariate Gaussian distribution** (see Eq. (3) and (4)).
>
> In Eq. (13), we introduce a linear Gaussian model for the target node, providing an **explicit expression for the conditional distribution** $p(f_X^{(i)}|pa_i)$ in Eq. (11). The corresponding multivariate Gaussian distribution, $p(f_X^{(i)})$, is derived in Proposition 3.1.
>
> We welcome the identification of any other unclear expressions that may pose challenges for understanding.
>
> >Q2. Additionally, a considerable portion of the explanations provided in Section 2 concerning directed graphs appears to be redundant. The authors delve into the intricacies of directed graphs and the definition of a Bayesian network, yet the second paragraph on page 4 **reveals that the graph relating the GPs in this paper is a fully connected undirected graph (as indicated by the definition of
>  for all output functions)**. It begs the question of why the authors did not emphasize this type of graph from the outset. There are several undirected graph models that can be pertinent when dealing with Gaussian variables, including Gaussian graphical models, pararnormal/nonparametric graphical models, and functional graphs.
>
> The definition of $pa_i$ **does not influence the graph type**; it solely designates the nodes with arrows pointing directly to the target node.
>
> In our proposed model, the **graph is directed** for the following reasons:
> 1. Two links exist between any two nodes due to the **asymmetry** of the interaction measured by the attention mechanism (with attention coefficients $\alpha_{ij} \neq \alpha_{ij}$).
> 2. Equation (11) defines a specific directed graph and establishes the fundamental basis for **expressing a multivariate Gaussian as a directed graph corresponding to a linear Gaussian model** (Equations (11), (12), and (13)).
>
> Furthermore, a **directed graphical representation is more fitting**, providing more flexible dependence measures. Additional elucidations have been incorporated in the updated version.
>
> >Q3. Indeed, the description of the distributed Gaussian process (DGP) in the paper is lacking. The conventional DGP model, as proposed by Deisenroth and Ng in 2015, operates **under two fundamental assumptions**. Firstly, it assumes that all GP experts share the same hyperparameters during the training phase, which serves as a mechanism to address overfitting. Secondly, it assumes perfect diversity between the target variable GPs. However, this second assumption does not align with the scope of MOGP, as **interactions between GPs are a crucial aspect** of the model. The paper also falls short of providing a clear explanation of how the distributed framework depicted in Figure 4 functions
>
> The proposed GMOGP framework can **seamlessly extend to distributed learning without resorting to extra approximations and assumptions**, as seen in the DGP. Figure 3 illustrates the fundamental framework distinctions between the GMOGP and standard MOGPs.
>
> In the GMOGP, each output has an independent workflow, leveraging shared interaction information from other outputs captured by kernel hyperparameters. Consequently, each output contributes specific information and amalgamates the information of dependent outputs. We have provided additional clarifications on this point and enhanced the descriptions in Figure 4 in the updated version.

---

> ### Author Response · Authors · 2023-11-20
> **Thanks for your review! please consider our response to your concerns - Part 2**
>
> >Q4. To mitigate computational costs, the model introduces **DGP, which can also be applied to conventional MOGP**. However, the paper falls short in discussing why this solution is faster than the other baselines. Both the attention mechanism and non-Gaussian transformation inherently raise computational costs, making it essential to have a comprehensive and detailed discussion on this issue within the paper. This is particularly important because it is one of the claims made in the paper. Regrettably, the authors only touch upon this matter in the experiments section, which is not sufficient to provide a thorough understanding of the computational efficiency of their approach.
>
> Directly applying the DGP to the conventional MOGP ($\mathcal{O}(N^3S^3)$) results in a **complete loss of interaction information** among other outputs. Consequently, the corresponding distributed MOGP **degrades to isolated SOGPs, rendering it meaningless**.
>
> In our experiments, we have already compared isolated SOGPs. The **key challenge lies in designing a distributed MOGP framework without sacrificing interaction information**, and the proposed GMOGP provides a viable solution.
>
> >Q5. A similar issue pertains to the prediction quality. The proposed solution integrates a combination of existing methods within a unified framework. However, the question arises: **why does this method enhance prediction quality and surpass other baselines**? While the experiments demonstrate a significant improvement, the underlying reasons have not been sufficiently elucidated within the paper.
>
> The proposed model outperforms others due to the following advantages:
> 1. **Enhanced Flexibility in Interaction Measures:** Unlike traditional approaches that measure dependence between outputs through covariance (kernel function), our model employs an attention mechanism. This introduces a more flexible and asymmetric dependence measurement, allowing for $\alpha_{ij}\neq \alpha_{ji}$.
> 2. **Separable Model Structure:** Our model eliminates the need to construct the large gram matrix $K_M\in\mathbb{R}^{SN\times SN}$ used in standard MOGPs. This streamlines computational and storage requirements.
> 3. **Efficient Learning Framework:** Diverging from the $S$-dimensional loss function typical in standard exact MOGPs (Equation 6), our model adopts a $1$-dimensional weighted sum objective (also the ELBO). This formulation facilitates the derivation of Pareto optimal solutions for kernel hyperparameters and distributed learning schemes.
> 4. **Additional representation ability:** The model can generate non-Gaussian prior/likelihood, alleviating the model misspecification on ubiquitous non-Gaussian data and providing graphic representations.
>
> See more details in Section 3; we added more description in the updated version.

---

> ### Comment · Reviewer_pYyL · 2023-11-22
>
> I appreciate the authors for their rebuttal and revised version of the paper. They addressed my concerns regarding the clarity of the paper, in particular about GMOGP workflow and directed graph structure. Since my points have been answered, I increased my score to 8.

---

### Official Review · Reviewer_sgU9 · 2023-10-30

**Soundness:** 3 good
**Presentation:** 4 excellent
**Contribution:** 3 good
**Rating:** 8
**Confidence:** 4

**Summary:**

The paper introduces a new construction of multioutput Gaussian processes (MOGP) by leveraging probability graphical models to capture dependencies among outputs through an attention mechanism. The Graphical MOGP (GMOGP) can be seen as a generalization of the classic MOGP-LMC, wherein dependencies (i.e., coefficients of coregionalization) are determined based on the output's parent nodes. The evidence lower bound of the GMOGP is introduced for the joint estimation (in a variational fashion) of the graphical structure and kernel parameters. The proposed framework is tested on several examples, and results allow assessing its competitivity w.r.t. the state-of-the-art.

**Strengths:**

A new construction of MOGP based on probability graphical models is introduced. The graphical model allows capturing dependencies through the definition of the parents of a given output. This makes it possible to define correlations between outputs that are not "bidirectional" (i.e. $\alpha_{i,j} \ne \alpha_{j,i}$ for $i \ne j$), which is not the case for the MOGP-LMC where a symmetric coregionalization matrix is required to promote invertibility. Both the graphical structure (attention coefficients) and kernel parameters are estimated in a variational context where the evidence lower bound of the GMOGP is introduced. The GMOGP inference involves less computational and storage ressources compared to the MOGP-LMC, allowing a better scaling of the model. In addition, the GMOGP can be performed in a distributed learning framework. The transformed GMOGP (TGMOGP) is introduced as well to deal with non-Gaussian priors/likelihoods.

The proposed GMOGP and TGMOGP are tested on several numerical examples, and results allow assessing their competitivity (and in many cases their superiority) w.r.t. other models from the state-of-the-art. Open-source codes based on pytorch/gpytorch are provided.

The paper is well-written and well-organized. The main theoretical developments are readable and notions well-motivated. The proposed discussions allows to easily understand how the GMOGP is placed w.r.t. the literature.

**Weaknesses:**

- The authors have claimed that their framework ensures pareto optimality but there is no numerical evidence. I suggest considering an example with only 2 outputs and adding plots allowing them to validate their claim. They can consider a well-known MOO toy example where the Pareto front has been already studied.
- Limitations of the proposed framework are not discussed throughout the paper. I believe that scaling the model to large datasets and/or to systems with tens/hundreds of outputs remain a challenge. I refer to **Questions** for further concerns.

**Questions:**

**Questions**

- Although the diversity of the examples, it is not completely clear that the model properly estimated the dependencies between the outputs. Can this be verified in a toy example? I suggest considering a toy example with 3 outputs where one of them is highly correlated to another, and the remaining one has no influence on the other ones. For instance, one may simulate $f_1, f_2$ from a GMOGP and $f_3$ from a SOGP. For the GMOGP, I suggest considering $\alpha_{1,2}^\ast \gg \alpha_{2,1}^\ast$. The other parameters of the graphical representation and kernels can be chosen arbitrarily. Once an illustrative sample is chosen, a new GMOGP can be trained considering data from the 3 outputs and initial values of cross-attention coefficients (non-diagonal elements) all equal (to 0.5). The framework is expected to recover the values of $\alpha_{1,2}, \alpha_{2,1}$ close to the ones used to generate the dataset and to estimate small values for $\alpha_{1,3}, \alpha_{2,3}, \alpha_{3,1}, \alpha_{3,2}$. It is also possible to compare the kernel parameters estimated by the GMOGP with the ones used to generate the data.
- When considering systems with a significant number of "potentially correlated" outputs (e.g. $S \geq 20$), one may think of promoting sparsity in the estimation of the attention coefficients. This may allow focusing the analysis on a subset of "highly correlated" nodes (outputs) while isolating the ones exhibiting a lesser influence on the others (i.e. attention coefficient close to zero). For instance, if $\alpha_{i,j} \approx 0$ for all $j = 1, \ldots, S$ and $j \ne i$ (there is no contribution to the other outputs $i$), and $\alpha_{j,i} \approx 0$ for all $i = 1, \ldots, S$ and $i \ne j$ (there is no relation with the other outputs $i$), then the output $j$ can be considered independent and it can be excluded from the graphical model. In the aforementioned example with 3 outputs, one may consider excluding the third output. Have the authors considered this experimental setup?
- Promoting sparsity in the attention coefficient values may simplify the complexity of the resulting model but implies overheads in the inference procedure since all the coefficients are jointly estimated. Is there a possibility to adapt the GMOGP inference scheme to sequentially estimate those coefficients? For instance, is it possible to start the GMOGP model with a minimal number of nodes (1-2) and add new ones using a criterion based on the level of the attention values?
- Can the authors provide further insights about the limitations of the proposed framework and potential solutions?

**Other minor remarks**
- Punctuation marks in the equations need to be double-checked throughout the paper (e.g. first line Eq.(3), expression after Eq.(6), Eq.(17)).
- Page 2, Section 2.1: "A Gaussian process..." $\to$ "A GP ..."
- Page 3, Footnote: To begin with capital letter
- Page 4, after Eq (12): $k_{\theta_i}$ is a **Co**variance function
- Page 5, after Eq (16): "In addition, Applying... "
- Page 6, Section 3.4: "1-d quadrature" $\to$ "1D quadrature"
- Page 9, Table 3: to display all the attention coefficients in the same table or to add pictures instead (see Figure 10b, Appendix B.4.1)
- In the references:
    - (Brody et al., 2021): already published in the ICLR 2022 (https://openreview.net/forum?id=F72ximsx7C1). The reference needs to be updated.
    - To promote uniformity in the references (styles, capital letters, names of the journals and conferences, ...).
    - (Hensmann et al., 2015): gaussian $\to$ Gaussian
    - Capital letters after ":"
    - (Williams and Rasmussen, 2006): the order of the authors is not correct.
    - (Velickovic et al., 2017): already published in International Conference on Learning Representations (https://openreview.net/forum?id=rJXMpikCZ). The reference needs to be updated.

---

> ### Author Response · Authors · 2023-11-20
> **Thanks for your thorough review! Response to Reviewer sgU9 (1/2)**
>
> We sincerely appreciate the time and effort you invested in reviewing our work. Your invaluable feedback has played a pivotal role in refining and improving the quality of our paper. We address your key concerns below.
> ***
> **Weaknesses**
> >W1. The authors have claimed that their framework ensures pareto optimality but there is no numerical evidence. I suggest considering an example with only 2 outputs and adding plots allowing them to validate their claim. They can consider a well-known MOO toy example where the Pareto front has been already studied.
>
> The objective function of the proposed GMOGP is $\sum_{i=1}^S w_i\mathcal{L}^{(i)}(\Theta)$. Given that kernel hyperparameters are shared across all nodes, optimizing for the kernel hyperparameters $\Theta$ corresponds to solving a Multi-Objective Optimization (MOO) problem. Utilizing weights greater than zero in the weighted sum method satisfies the conditions for a Pareto optimal solution of $\Theta$, as detailed in the paper (Marler & Arora, 2010). The inequality constraint in the primal MOO problem serves as a valid kernel constraint.
>
> However, it's crucial to emphasize that our weighted sum loss function still **relies on parameters within the attention mechanism and noise variance**. These learning parameters are not shared among outputs but are tailored to **simultaneously minimize all objectives to zero**.
> This clarification is detailed in the updated version.
>
> >W2. **Limitations** of the proposed framework are not discussed throughout the paper. I believe that scaling the model to **large datasets** and/or to systems with **tens/hundreds** of outputs remain a challenge. Can the authors provide further insights about the limitations of the proposed framework and potential solutions?
>
> For large datasets, we refer to using the TGMOGP, which is learned by sparse variational inference. For tens/hundreds of outputs, one can use the distributed (T)GMOGP, but the limitation goes with the hardware.
>
> Key limitations of the GMOGP can be mitigated through:
> 1. **Interpretation of Attention Coefficients:** Explore insights into attention coefficients learned by TGMOGP and leading graph attention networks.
> 2. **Dynamic Dependence Measures:** Adapt to changing interactions by exploring dynamic dependence measures.
> 3. **Extensions for Online Learning and Graph Construction:** Enhance real-time adaptability with online learning and dynamic graph construction extensions.
> 4. **Tailored Attention Mechanisms/Scoring Functions:** Optimize performance by tailoring attention mechanisms, scoring functions, or dependence measures.
> 5. **Alternative Aggregation Models** (Eq. 13): Explore diverse aggregation models for capturing intricate relationships.
> 6. **Custom Objective Functions (Weights Setting):** Tailor objectives and weights to diverse applications for enhanced flexibility.
>
> These strategies present opportunities for refining and expanding the proposed model.
> ***
> **Questions**
> >Q1.  Although the diversity of the examples, it is not completely clear that the model properly estimated the dependencies between the outputs.
>
> We simulated 200 samples of $f_1$ and $f_2$ from a GMOGP with $\alpha_{12} = 0.5$ and $\alpha_{21} = 0.8$, and $f_3$ from a SOGP. The kernels are SE with length scales $\theta_1 = 3$, $\theta_2 = 5$, and $\theta_3 = 0.5$ respectively.
> Using Adam with a learning rate of 0.08, after 500 iterations, the recovered interaction attention coefficients are $\alpha_{12} = 0.8811$, $\alpha_{21} = 0.9810$, $\alpha_{13} = 0.0086$, $\alpha_{23} = 0.0086$, $\alpha_{31} = 0.2311$, $\alpha_{32} = 0.2327$. The estimated kernel hyperparameters are $\theta_1 = 3.044$, $\theta_2 = 3.245$, and $\theta_3 = 0.295$. The results imply that output$_1$ and output$_2$ are highly correlated. However, identical estimates of the underlying parameters occur only when the kernel hyperparameters have the same values. The problem is analogous to a linear system with three variables (undefined coefficients) and only one equation, providing non-unique solutions.

---

> > ### Author Response · Authors · 2023-11-20
> > **Thanks for your thorough review! Response to Reviewer sgU9 (2/2)**
> >
> > >Q2. When considering systems with a significant number of "potentially correlated" outputs (e.g. $S\ge20$), one may think of promoting sparsity in the estimation of the attention coefficients. This may allow focusing the analysis on a subset of "highly correlated" nodes (outputs) while isolating the ones exhibiting a lesser influence on the others (i.e. attention coefficient close to zero). ... In the aforementioned example with 3 outputs, one may consider excluding the third output. Have the authors considered this experimental setup?
> >
> > Precisely, our GMOGP can select the highly correlated outputs (parents).
> >
> > Since we lack graph knowledge, we initially consider all other outputs as potential parents of the target node $i$. During the training procedure, we unlink the outputs with coefficients $\alpha_{ij} \approx 0$ (as discussed in Section 3.2). Table 3, based on the SARCOS dataset, illustrates that the parents set of the second output only includes the node corresponding to the 4th output. This is evident as $\alpha_{21} \approx 0$, $\alpha_{23} \approx 0$, and $\alpha_{24}$ is large. For each target output, we can select different contributing parents. Figure 3 visually demonstrates that each output only updates the corresponding learning parameters with the selected parents.
> >
> > If both $\alpha_{ij} \approx 0$ and $\alpha_{ji} \approx 0$ simultaneously, the GMOGP with respect to output $i$ regresses to an isolated GP. In this scenario, it only updates its own parameters and can be treated as an isolated node without edges. However, we still need to perform the regression task for output $i$. Otherwise, we can discard this node.
> >
> > >Q3. Promoting **sparsity** in the attention coefficient values may simplify the complexity of the resulting model but implies overheads in the inference procedure since all the coefficients are jointly estimated. Is there a possibility to adapt the GMOGP inference scheme to **sequentially estimate** those coefficients? For instance, is it possible to start the GMOGP model with a minimal number of nodes (1-2) and add new ones using a criterion based on the level of the attention values?
> >
> > Isolating one or several nodes does not significantly impact the overall complexity since we aim to predict all nodes. Additional complexity reduction can be attained through covariance matrix approximation, sparse/distributed learning and inference, etc.
> >
> > For the sequential estimation, it depends on the task. If the goal is solely to predict node 1, and we have already collected data from node 2 (with estimated $\alpha_{12}, \boldsymbol{\theta}_1, \boldsymbol{\theta}_2$),
> >
> > one can fix these parameters and focus solely on learning $\alpha_{13}$ for the new node 3
> > (where $\boldsymbol{\theta}_3$ needs to be estimated by an SOGP initially).
> >
> > Subsequently, the
> > GMOGP prior for node 1 is updated by incorporating $\alpha_{13} \boldsymbol{\theta}_3$.
> >
> > Various improvements and GMOGP variants corresponding to different tasks can be considered. The revised version includes additional experiments, limitations, and extra descriptions, with corrections for typos and errors.
> >
> > > Q4. Can the authors provide further insights about the limitations of the proposed framework and potential solutions?
> >
> > See the response to W2.

---

> > ### Comment · Reviewer_sgU9 · 2023-11-20
> >
> > I thank the authors for answering my queries. Concerning the numerical example considering $f_1$ and $f_2$ from a GMOGP and $f_3$ from a SOGP, is there any improvement (i.e. estimation of $\alpha_{i,j}$ for all $i, j = 1,2,3$ close to the true ones) when the number of samples increases? If not, can the authors justify the reason for having $\alpha_{3,1}$ and $\alpha_{2,3}$ not very close to zero (knowing that $f_3$ is an independent SOGP)?

---

> > > ### Author Response · Authors · 2023-11-21
> > > **Proposing an efficient and flexible exact MOGP framework. Reply to Reviewer sgU9.**
> > >
> > > Thank you for your feedback, we appreciate the time and effort you dedicated to evaluating our work. We sincerely hope that our answers have fully addressed your concerns.
> > > ***
> > > In general, the recovering problem involves generating samples from a GP using specific kernels, expressed as $k = k_{\theta_1} + ak_{\theta_2}$. Subsequently, another GP is utilized to fit these samples with the same kernel form, $\hat{k} = k_{\theta_3} + bk_{\theta_4}$. The objective is to ensure that, given the same inputs, the kernel function values are identical, i.e., $ k = \hat{k}$. While various hyperparameter sets can satisfy this equation, an exact match requires $\theta_1 = \theta_3$ and $\theta_2 = \theta_4$, leading to $ b = a $.
> > > In practice, employing isolated GPs for pretraining each output's samples offers a beneficial strategy, providing a robust initial guess for kernel hyperparameters.
> > >
> > > For the specific example with three outputs, despite $f_3$ being sampled from a SOGP, the Pearson correlation coefficients between $f_1$ and $f_3$ (0.1073) and between $f_2$ and $f_3$ (0.2058) indicate some degree of interdependence. The non-zero values of $\alpha_{31}$ and $\alpha_{32}$ can be attributed to $ \theta_3 = 0.295$ being smaller than the underlying one (0.5), necessitating compensation from other nodes. Changing the underlying kernel of $f_3$ to a non-isotropic linear kernel results in all attention coefficients with respect to $f_3$ converging to zero.
> > >
> > > In this paper, our main objective is to enhance the predictive performance through an efficient MOGP framework.

---

> > > > ### Comment · Reviewer_sgU9 · 2023-11-22
> > > >
> > > > I thank the authors for these clarifications. Since they have correctly addressed my concerns (and most of those of the other reviewers), I have decided to increase my rating to 8 (accepted, good article).

---

### Official Review · Reviewer_9P5Q · 2023-10-31

**Soundness:** 3 good
**Presentation:** 4 excellent
**Contribution:** 3 good
**Rating:** 8
**Confidence:** 4

**Summary:**

This work proposes a multi-output regression framework, termed graphical MOGP (GMOGP). The GMOGP is built upon a probability graphical model that can learn the conditional dependence and imply graphical  representations of the multiple outputs. The framework allows learning of the model hyper-parameters via a multi-objective optimization problem. The work also provides an extension by non-linear transformations in order to fit non-Gaussian data.

**Strengths:**

Originality: To the best of my knowledge the work seems original, built upon different ideas from the state-of-the-art that the authors appropriately merged to compliment each other.

Quality: The experiments, both with synthetic and real world data, are useful to provide an idea of the approach.

Clarity: The work is very well written in general with neat notations and explanations of the work. Quite consistence in the mathematical notation. There were just a few typos to easily fix.

Significance (importance): The application of Multioutput Gaussian processes tend to be prohibited in the practice when the number of data observations and  outputs grows. Therefore this work is significant due to allowing the MOGP computational complexities to be considerably reduced and additionally permitting the non-conjugate likelihood/prior scenario to be applied in a practical manner.

**Weaknesses:**

The experiments, both with synthetic and real world data, are useful to provide an idea of the approach. Nonetheless, the work lacked of discussion, citation and experimentation of the Gaussian process models based on convolution processes, a more general and powerful approach than the Linear Model of Coregionalisation (LMC).

There is not discussion in the paper about the limitations of the work.

**Questions:**

---Specific comments---

-In the Introduction, where it reads "the input-output relationship with multivariate Gaussian...", better "...with a multivariate Gaussian..."

-I did not see any literature related to MOGPs with convolution processes which might indeed be a more general approach than the LMC. See for instance works like:

"P. Boyle and M. Frean, Dependent Gaussian processes (2005)", "J-J. Giraldo et al, Correlated Chained Gaussian Processes for Modelling Citizens Mobility Using a Zero-Inflated Poisson Likelihood (2022)", "T. M. McDonald et al, Nonparametric Gaussian Process Covariances via Multidimensional Convolutions (2023)", "M. A. Alvarez and N. D. Lawrence, Computationally Efficient Convolved Multiple Output Gaussian Processes (2011)"

-I do not follow what the authors mean by "on a set of shard GPs". Is shard an adjective? Was it shared GPs maybe?

-In section 2: "...is parametrized by: length-scale $l$ and signal variance...", better to use "...by: a length-scale $l$ and a signal variance..."?

-After Eq. (3), change "vector-valued function... are determined by...", by "vector-valued function... is determined by..."

-After Eq. (4), with the current notation $Y$ might be understood as a matrix instead of a vector, it might be better to clarify the dimensionality of $Y = [{\mathbf{y}^{(1)}}^\top,...,{\mathbf{y}^{(i)}}^\top,...{\mathbf{y}^{(S)}}^\top]^\top \in \mathbb{R}^{NS}$.

-After Eq. (7), shouldn't it be $K_{i,i^{\prime}}(\mathbf{x},\mathbf{x}^{\prime})=\sum_{q=1}^{Q}a_{iq}a_{i^{\prime}q}k_q(\mathbf{x},\mathbf{x}^{\prime})$?

-In section 3.3, lower case "applying" in "In addition, Applying". Also, include "a Gaussian" for "...follows Gaussian that can be normalized..."

-After Eq. (18), the functions $f_1(x), f_2(x),f_3(x),f_4(x)$ should receive a vector $\mathbf{x}=[x_1,x_2]^\top$ as argument, i.e., $f_1(\mathbf{x}), f_2(\mathbf{x}),f_3(\mathbf{x}),f_4(\mathbf{x})$.

-In the figure 5(c), correct the legend "GRPN" to "GPRN".

---Other Questions---

-In the GMOGP I only noticed a derivation for isotopic data, but I did not see any mention that the method would allow the use of heterotopic data, would the method allow its application to heterotopic data?

-In line with the last question, how is the behaviour of the GMOGP in the context of notable unbalanced data per output?

-It is mentioned that the weights $w_i$ are set equal since there is not priority among the outputs, is this assumption still valid in the context of markable unbalanced data along the outputs?

-From the practitioner perspective, what would it be an appropriate way to initialise the weights $w_i$? Is there much effect on initialising them quite small (say $(0,1]$) or quite big ($>10$)?

-It might be useful to also show the performance comparison with a MOGP with Convolution processes a more general and powerful approach than the LMC.

It would be important to also discuss about the possible limitations of the work, there was nothing related to it in the paper.

---

> ### Author Response · Authors · 2023-11-20
> **Appreciating your valuable Feedback! Response to Reviewer 9P5Q**
>
> We sincerely appreciate your valuable feedback! Your insights greatly contribute to enhancing the quality of our work. Additional experiments and clarification are carefully refined. We address your key concerns below.
> ***
> **Weaknesses**
> >W1. Nonetheless, the work lacked of discussion, citation and experimentation of the Gaussian process models based on convolution processes, a more general and powerful approach than the Linear Model of Coregionalisation (LMC).
>
> We included experiments for the convolutional MOGP (CMOGP (M. A. Alvarez and N. D. Lawrence, 2011)), a non-separable model, which outperforms most baseline MOGPs. The results show that the CMOGP outperforms most baseline MOGPs.
> |  | SOGP | CMOGP|V-LMC |SGPRN |FICM |GMOGP|TGMOGP|
> |------------------|------------------|------------------|------------------|------------------|------------------|------------------|------------------|
> | Synthetic data  | 0.5653$\pm$0.0023 | 0.5539$\pm$0.0089  |0.5917$\pm$0.0096   |0.5819$\pm$0.0207 |0.5544$\pm$0.0046 |0.5541$\pm$0.0054  |0.5343$\pm$0.0023 |
> | Jura   | 0.605$\pm$0.01|0.429$\pm$0.01 |0.443$\pm$0.01 |0.438$\pm$0.02 |0.394$\pm$0.05| 0.376$\pm$0.01 |0.382$\pm$0.01  |
> ***
> In the revised version, we have added experiments, limitations, and additional descriptions. Typos and errors have been rectified.
>
> >W2. There is not discussion in the paper about the limitations of the work.
>
> The main limitations of the GMOGP can be mitigated through various avenues:
> 1. **Interpretation of Attention Coefficients**: The attention coefficients signify dependencies among nodes and graph structures in this paper. However, interpretations of the coefficients learned by the TGMOGP and state-of-the-art (SOTA) graph attention networks are anticipated in Explainable Artificial Intelligence (XAI).
> 2. **Dynamic Dependence Measure**: The interaction or dependence dynamically changes with inputs.
> 3. **Extensions of Online Learning Frameworks and Graph Construction Schemes**: Opportunities exist for extending the GMOGP to online learning frameworks and enhancing graph construction schemes in an online setting.
> 4. **Adaptation of Attention Mechanism/Scoring Function/Dependence Measure** for Different Tasks: Consideration of diverse attention mechanisms, scoring functions, and dependence measures tailored to specific tasks.
> 5. Exploration of **Different Aggregation Models** (Eq. (13)): Investigating alternative aggregation models, as defined in Equation (13).
> 6. Exploration of **Different Objective Functions (Weight Settings)** for Varied Applications: Exploring diverse objective functions and weight settings tailored to different applications.
>
>
> ***
> **Questions**
> >Q1&Q2. In the GMOGP I only noticed a derivation for isotopic data, but I did not see any mention that the method would allow the use of heterotopic data, **would the method allow its application to heterotopic data?** In line with the last question, how is the behaviour of the GMOGP in the context of notable **unbalanced data per output**?
>
> The proposed GMOGP can handle heterotopic data ($X^{(1)}\neq… \neq X^{(S)}$). For simplicity, we use the same notation $X$. This point has been clarified in the updated version.
>
> In the case of unbalanced data, where different sample sizes exist, an additional weight matrix is required in the attention mechanism to map the inputs of Equation (15) to the same dimensions. This weight matrix is commonly applied in graph attention networks. Instead of the inner product measure in the scoring function (Equation (15)), other types of scoring functions such as concatenation measure can be considered for unbalanced data.
>
> The GMOGP prior **remains consistent** in the context of unbalanced data, as it depends only on its own training data samples, and the kernel hyperparameters are shared to exchange information among outputs. However, it's noted that the specific information in the output with a small sample size may be **inadequately captured**. Descriptions addressing this point have been added in the updated version.
>
> >Q3. It is mentioned that the weights $w_i$ are set equal since there is not priority among the outputs, is this assumption still valid in the context of markable unbalanced data along the outputs?
>
> If we aim to predict all outputs simultaneously, it is preferable to assign equal weights to the loss function for each output, regardless of the number of samples provided. In cases where certain outputs are associated with low-quality data (biased labels), consideration should be given to assigning a lower weight.
>
> >Q4. From the practitioner perspective, what would it be an appropriate way to initialise the weights $w_i$?
>
> In our experiments, we set all weights to 1 since there is no preference for the outputs. Practical guidelines for setting weights are outlined in the paper by (Marler & Arora (2010))
>
> >Q5 ... the performance comparison with a MOGP with Convolution processes ...
>
> Refer to W1.
>
> >Q6 ...limitations...
>
> Refer to W2.

---

> > ### Comment · Reviewer_9P5Q · 2023-11-22
> >
> > Thanks to the authors for addressing the different comments and questions of the paper. There has been an additional improvement of the work which makes it an important contribution to the research community.

---

### Author Response · Authors · 2023-11-22
**Summary of revisions made to the paper in the discussion period**

We extend our gratitude to the reviewers for their valuable feedback. We acknowledge and appreciate the insightful comments provided. We agree that the clarifications strengthen the support for the proposed model, resulting in a stronger paper.
***

In this revised version (2023-11-22), we have addressed the following points as raised by the reviewers:
* Emphasized the flexible asymmetric dependence measure in our GMOGP (Section 3.2, as asked by Reviewer X7C1).
* Clarified the GMOGP workflow, and explained the revised Figure 3 and Figure 4 in the main text and caption (as required by Reviewer pYyL and X7C1).
* Clarified the distinctions between our distributed GMOGP and the classic distributed Gaussian process (DGP) (Section 3.3 and Remark 3.2, as requested by Reviewer pYyL).
* Clarified the “enlarged searching space...” and "non-linear effect" among outputs (Section 3.3 & 3.4, as asked by Reviewer X7C1).
* Emphasized the directed graph structure (see Figure 2 and Appendix A, as requested by Reviewer pYyL).
* Clarified the Pareto optimal solutions of kernel hypers and parent selection strategy (Section 3.2, as asked by Reviewer sgU9).
* Included experiments of MOGP with the convolutional process and heterotopic data analysis. Limitations and possible extensions are summarized in the Appendix.
* Corrected all typos and errors (as requested by Reviewer 9P5Q and sgU9).


We believe that we have addressed the questions and concerns raised by the reviewers. If they find the revisions satisfactory, we hope that they would consider raising their score. If any questions or concerns remain, we are more than willing to engage in further discussion.


Recap: To enhance the predictive performance of MOGPs in terms of flexibility, optimality, and scalability, the proposed GMOGP is fortified by:

1. Flexible dependence measure (asymmetric);
2. Generating flexible Gaussian process priors consolidated from identified parents;
3. Providing dependent processes with attention-based graphical representations;
4. Achieving Pareto optimal solutions of kernel hyperparameters via exact distributed learning.


Please feel free to reach out if you have any further questions or concerns. We are open to additional discussions or addressing any further comments you may have.

---

### Author Response · Authors · 2023-11-23
**General Response to All Thoughtful Reviewers**

We extend our sincere gratitude to all the reviewers for their time, careful evaluation, and positive adjustments. Your collective support is invaluable, and we are truly grateful for your significant contributions to the review process.

We are committed to addressing any remaining concerns to further enhance the quality of our paper. If you have any additional feedback or questions, please do not hesitate to share them.

---

### Meta-Review · Area_Chair_Z79p · 2023-12-05

**Metareview:**

The reviewers found the proposed approach for multi-output regression original and well presented, with good empirical evaluation and examples. The paper was subject to good comments and discussion during the discussion phase. In the end, all reviewers appreciated the contributions of this paper and all four reviewers recommended accepting this work to ICLR.

**Justification For Why Not Higher Score:**

The paper is interesting, but maybe not topical-enough for an oral.

**Justification For Why Not Lower Score:**

The paper could be downgraded to a poster if needed.

---

### Decision · Program_Chairs · 2024-01-16

Accept (spotlight)